# Enhancing Temporal Understanding in Video-LLMs through Stacked Temporal Attention in Vision Encoders

**Ali Rasekh**
Leibniz University Hannover,
L3S Research Center
ali.rasekh@L3S.de

**Erfan Bagheri Soula**[*]
Independent Researcher
erfan.b.soula@gmail.com

**Omid Daliran**[*]
Independent Researcher
hopebraves@gmail.com

**Simon Gottschalk**
Leibniz University Hannover,
L3S Research Center
gottschalk@L3S.de

**Mohsen Fayyaz**
Microsoft
mohsenfayyaz@microsoft.com

## Abstract

Despite significant advances in Multimodal Large Language Models (MLLMs), understanding complex temporal dynamics in videos remains a major challenge. Our experiments show that current Video Large Language Model (Video-LLM) architectures have critical limitations in temporal understanding, struggling with tasks that require detailed comprehension of action sequences and temporal progression. In this work, we propose a Video-LLM architecture that introduces stacked temporal attention modules directly within the vision encoder. This design incorporates a temporal attention in vision encoder, enabling the model to better capture the progression of actions and the relationships between frames before passing visual tokens to the LLM. Our results show that this approach significantly improves temporal reasoning and outperforms existing models in video question answering tasks, specifically in action recognition. We improve on benchmarks including VITATECS, MVBench, and Video-MME by up to +5.5%. By enhancing the vision encoder with temporal structure, we address a critical gap in video understanding for Video-LLMs. Project page and code are available at: https://alirasekh.github.io/STAVEQ2/

## 1 Introduction

Recent advances in Multimodal Large Language Models (MLLMs) have led to significant improvements in performance across a wide range of tasks involving multimodal data, including video question answering and image captioning – demonstrating impressive capabilities in integrating both visual and textual information. However, despite these advancements, when it comes to videos, current Video Large Language Models (Video-LLMs) still face major challenges in understanding temporal dynamics.

In video question answering (VQA), the input is a video accompanied by a natural language question, and the output is a textual answer to that question. While current models perform reasonably well on spatially grounded questions (e.g., "What color is the ball?"), they struggle when the query requires precise comprehension of temporal progression within the video. Notably, many existing benchmarks

---

[*]Denotes equal contribution.

39th Conference on Neural Information Processing Systems (NeurIPS 2025).

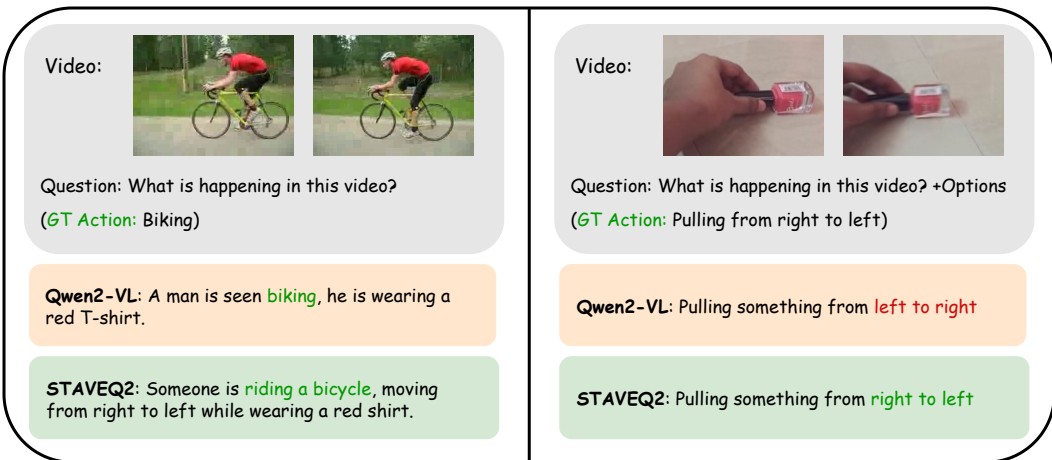

Figure 1: Responses from Qwen2-VL and our STAVEQ2. **Left:** For a temporally simple action (Biking), both models answer correctly. **Right:** For a temporally challenging action (pulling something from right to left), Qwen2-VL provides an incorrect answer, while our STAVEQ2 succeeds.

and datasets are not particularly challenging in terms of temporal complexity, allowing some models to answer questions by relying on a single frame. In contrast, tasks such as action recognition necessitate understanding not only frame content but also how the frames change over time. An example is given in Figure 1, where the action captured in the second video is identifiable only when observing at least two frames.

In this work, we first analyze the limitations of current Video-LLMs such as Qwen2-VL [44] and InternVideo2-Chat [47] in temporal modeling capabilities. Specifically, we observe that current Video-LLMs struggle with fine-grained temporal reasoning tasks, such as distinguishing between actions that differ subtly in their execution over time (e.g., pulling an object from left to right versus right to left). Furthermore, despite attempts to improve performance through in-context learning approaches, these models consistently fail to recognize similar temporal patterns across different video instances, indicating a fundamental limitation in their temporal processing architecture rather than merely a training data issue. Building upon our findings in the limitations of the current state-of-the-art models, we propose our novel video-LLM model STAVEQ2.

With STAVEQ2, we propose an enhanced Video-LLM architecture that introduces stacked temporal attention modules directly within the vision encoder. As detailed in Figure 3, this architectural change explicitly equips a Video-LLM's vision encoder with temporal attention, enabling it to better capture the progression and dynamics of actions across frames before passing the visual tokens to the LLM for final reasoning. In our evaluation, we demonstrate that this design significantly improves the model's temporal understanding, leading to superior performance on temporally challenging tasks and benchmarks.

In summary, our contributions include: (i) We analyze how current Video-LLMs struggle in capturing complex temporal dynamics, even with in-context examples and fine-tuning. (ii) We propose STAVEQ2, our Video-LLM model, equipped with improved temporal video understanding. We show that STAVEQ2 outperforms recent state-of-the-art Video-LLMs on several benchmarks. To the best of our knowledge, with STAVEQ2, we are the first to efficiently include dedicated temporal attention blocks into the vision encoder of Video-LLMs for video question answering. (iii) We achieve new state-of-the-art results on the SSv2 action recognition benchmark by applying our proposed temporal attention mechanism to previous state-of-the-art video foundation model (Vision-only).

## 2   Related Work

**Video-LLMs**   The field of video understanding has witnessed significant advancements with the emergence of Video-LLMs. Multimodal language models, which integrate additional modalities like images [42, 31, 37, 20] or audio [11, 23, 8] into language models, have expanded the scope of LLMs beyond text [12]. Extending this to the video domain, recent Video-LLMs have enabled tasks such as

video captioning, question answering, and instruction following by aligning video content with natural language. Models like Video-ChatGPT [33], Video-LLaMA[53], Video-LLaVA [30], LLaVA-NeXT-Video[54], VideoChat [27], Otter [26], Gpt4Video [49], and InternVL 2.5 [9] demonstrate strong performance in grounding text in visual inputs. While they effectively capture spatial semantics, many still struggle with modeling long-range temporal dependencies and maintaining consistency across frames. This challenge is underscored in some works [2], which show case that even state-of-the-art video-language models lack an inherent understanding of temporal order, and proposes additional temporal information is necessary to improve these lacks. Our work builds on these models by focusing on improving temporal understanding in video-based reasoning tasks.

**Temporal Understanding in Video-LLMs**  Despite the advancements in Video-LLMs, temporal understanding is still an area that has not been fully explored, and unlike claims provoked by Liu et al. [32], existing models appear to struggle with understanding temporally complex tasks. Several works have made efforts to address this, such as those by [25, 35, 22], specifically targeting very long videos [41, 40, 24, 18]. Also, some works have been trying to push temporal understanding on models using methods such as temporal localization or boundary [6, 35, 22]. Others use Q-formers to condition the temporal features on inputs in order to improve temporal understanding and performance [36, 1]. TG-Vid [21] introduces a time gating mechanism aimed at enhancing temporal modeling; however, its results remain modest and the method is not particularly efficient.

Prior works employ varied approaches to temporal modeling. For instance, Qwen2-VL relies solely on spatial attention in its vision encoder, delegating temporal understanding to the language model. In contrast, InternVideo2 incorporates joint spatiotemporal attention in its vision encoder. However, our experiments in the next section demonstrate that both approaches are insufficient for capturing fine-grained motion dynamics and temporal dependencies. However, another approach that has been less explored in Video-LLMs is the divided space-time attention introduced in [5]. Our experiments also explore multimodal in-context learning in VLMs, as studied in [16, 7, 38], revealing that existing models lack sufficient in-context learning capability.

## 3 Problem Definition & Motivation

Before introducing our model, we begin by analyzing the performance limitations of current Video-LLMs on temporally demanding video question answering (VQA) tasks. Through a series of experiments, we reveal fundamental weaknesses in temporal reasoning, which motivate the need for our proposed approach.

**Problem Definition**  We address the task of video question answering (VQA), where the input consists of a video $\mathcal{V} \in \mathbb{R}^{T \times H \times W \times 3}$ with $T$ RGB frames of height $H$ and width $W$, and a natural-language question $\mathcal{Q}$. The goal is to produce a textual answer $\mathcal{A}$ that accurately responds to the question based on the visual content of the video. This formulation generalizes to a variety of tasks such as video captioning, action recognition, and similarity detection. For example, action recognition can be cast as a VQA task by posing the question: "What action is happening in this video?" with an answer such as "Pulling [something] from left to right." Our focus is on temporally challenging VQA tasks, where the correct answer depends not on static frames, but on modeling the sequence and progression of visual elements over time. For instance, distinguishing an action such as moving an object from left to right versus from right to left requires precise temporal reasoning.

We create a temporally challenging VQA dataset, specifically curated to stress temporal understanding. We evaluate the performance of selected Video-LLMs (Qwen2-VL, InternVideo2-Chat, and LLaVA-NeXT-Video; see Section 2) on this dataset under zero-shot settings. Then, we investigate whether performance can be improved via conventional LLM improvement approaches such as through in-context learning. Finally, we analyze the results to demonstrate the inherently temporal nature of our dataset and the persistent limitations of current models in handling such tasks.

### 3.1 Temporally Challenging Dataset

Our experiments utilize the Something-Something v2 (**SSv2**) dataset [17, 34] – an action recognition benchmark with over 220K videos, each annotated with one of 174 actions (e.g., "Pulling [something] from left to right"). To examine temporal challenges, we select a subset of SSv2 with action classes

Table 1: Zero-shot and in-context action recognition performance of four Video-LLMs. We use a different LLM to judge whether the generated answer matches the ground-truth action (LLM-as-a-Judge). *We provide the 10 possible actions within the prompt.

| # Examples | Qwen2-VL 2B | Qwen2-VL 7B | InternVideo2 Chat 8B | LLaVA NeXT-Video 7B |
|---|---|---|---|---|
| 0 | 14.87% | 21.91% | 30.60% | 19.38% |
| 0* | 24.01% | 35.91% | 46.11% | 31.46% |
| 1 | 9.24% | 15.83% | 23.86% | 16.54% |
| 3 | 9.56% | 16.79% | 17.04% | 18.98% |
| 5 | 8.92% | 20.72% | 10.41% | 16.32% |

that are opposites in how they change over time. This way, we create the dataset **SSv2-T10**, that specifically targets at temporally-challenging cases by reducing SSv2 to 14,462 videos of 10 classes that represent pairwise counterparts regarding their temporal characteristics. These classes were chosen to create temporally challenging VQA tasks, where accurate action recognition depends on modeling the sequence and progression of visual elements across frames, aligning with our goal of evaluating Video-LLMs' temporal reasoning capabilities. The list of selected action classes is provided in Appendix B.

## 3.2 Zero-Shot & In-Context Performance

We analyze whether the performance of the tested Video-LLMs can be improved through few-shot prompting, motivated by findings in multimodal LLMs showing that in-context learning enhances performance [14, 52]. As shown in Table 1, they perform poorly in the pure zero-shot setting — with accuracies as low as 14.87% for Qwen2-VL 2B and only 30.60% for InternVideo2-Chat. Even when we provide the list of 10 candidate classes within the prompt, performance remains weak. Although scores improve notably in this setting (e.g., 46.11% for InternVideo2-Chat and 35.91% for Qwen2-VL 7B), the models still often fail to select the correct class, despite knowing the task and being given the exact set of possible answers. This indicates a limited understanding of the temporal aspects of video content. Such a weakness is critical, as one of the most important cues in video-based tasks—namely, the direction and progression of motion—is not reliably captured.

To improve performance, we provide examples of videos and their actions in the respective prompts (see Appendix A for example prompts). However, the second part of Table 1 demonstrates that this strategy does not help. In fact, all models show performance drops as examples are added. Notably, all models except Qwen2-VL 7B drop in performance when the number of examples increase to 5. These results suggest that current video-language models struggle with in-context learning for VQA, failing to recognize actions across video instances even when given explicit examples. We provide more experiments on different settings in Appendix F.

## 3.3 Temporally Challenging VQA

In this experiment, we focus on InternVideo2-Chat, as it achieves the best zero-shot performance among the evaluated models–likely due to its spatiotemporal attention mechanisms in the vision encoder. Figures 2a and 2b present confusion matrices for InternVideo2-Chat on SSv2-T10, treating action recognition as a classification task. Without fine-tuning, the model struggles to distinguish between directional actions such as "Pulling [something] from left to right" versus "Pulling [something] from right to left", which are different in their temporal aspects. After fine-tuning on SSv2-T10, its performance improves, but significant confusion between temporally mirrored actions remains, it still fails to reliably distinguish actions that have different temporal meanings. These are actions that involve the same objects, actions and context and are visually very similar, but differ only in their temporal direction or order.

This result highlights a critical limitation: although InternVideo2-Chat is architecturally better equipped for temporal reasoning than other models, it still fails to reliably model directional information, a fundamental aspect of video understanding. We attribute this to the lack of dedicated temporal

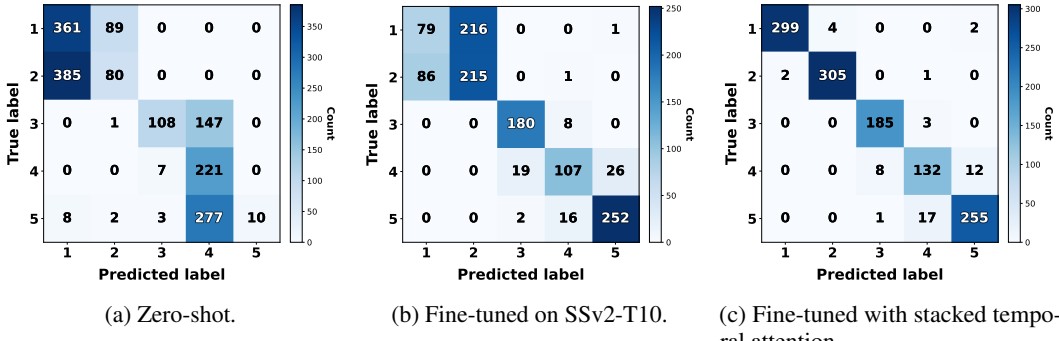

(a) Zero-shot.

(b) Fine-tuned on SSv2-T10.

(c) Fine-tuned with stacked temporal attention.

Figure 2: Confusion matrices of InternVideo2-Chat performing action recognition on SSv2-T10 showing results on the following classes: (1) Pulling [something] from left to right; (2) Pulling [something] from right to left; (3) Throwing [something] in the air and catching it; (4) Throwing [something] in the air and letting it fall; (5) [Something] falling like a rock.

modeling blocks in current Video-LLMs, which hinders their ability to reason about fine-grained temporal structures, even when spatiotemporal cues are present in the architecture. As shown in Figure 2c, after adding stacked temporal blocks, we can see that the model performs much better and is able to distinguish the actions accurately. This improvement confirms the importance of explicit temporal modeling in Video-LLMs. Corresponding quantitative results of this experiment are provided in Appendix D (Table 8)

## 4 Model: STAVEQ2

Through our initial experiments in the previous section, we demonstrated the difficulties of state-of-the-art Video-LLMs to deal with temporally challenging VQA tasks, even when in-context examples were provided. To address these challenges, we introduce our approach for enhancing temporal understanding in Video-LLMs in Video-LLMs: STAVEQ2 – Stacked Temporal Attention in Visual Encoders for Qwen2-VL.

### 4.1 Preliminaries

MLLMs typically combine a vision encoder with an LLM to process visual and textual inputs [31]. The vision encoder extracts features from visual inputs (e.g., images or videos), which are then transformed into token embeddings via trainable adapters. These embeddings are fed into the LLM alongside textual inputs, enabling joint visual and linguistic understanding. For Video-LLMs, the vision encoder must effectively capture both spatial and temporal information to model the dynamics of video sequences.

Given a video $\mathcal{V}$, following the vision transformer paradigm [13], a vision encoder typically divides each frame into $N = HW/P^2$ non-overlapping patches of size $P \times P$. These patches are projected into an embedding space, yielding a sequence of patch embeddings $X^{(0)} \in \mathbb{R}^{T \times N \times D}$, where $X_{t,i}^{(0)} \in \mathbb{R}^D$ denotes the embedding of the $i$-th patch in frame $t$, and $D$ is the embedding dimension. Each transformer block in the vision encoder applies multi-head self-attention across the $N$ patches within each frame, followed by a feed-forward multilayer perceptron (MLP), producing spatially informed feature representations.

### 4.2 Overview

Our experiments show that relying entirely on the LLM to interpret temporal relationships between frames as by Qwen2-VL is insufficient. Even the use of joint spatiotemporal attention in models like InternVideo2-Chat cannot adequately resolve this issue, as shown in our temporal analysis (Section 3.3). Furthermore, InternVideo2-Chat is architecturally constrained to processing 8 input frames, which limits its applicability to short-video tasks and precludes its use on longer video

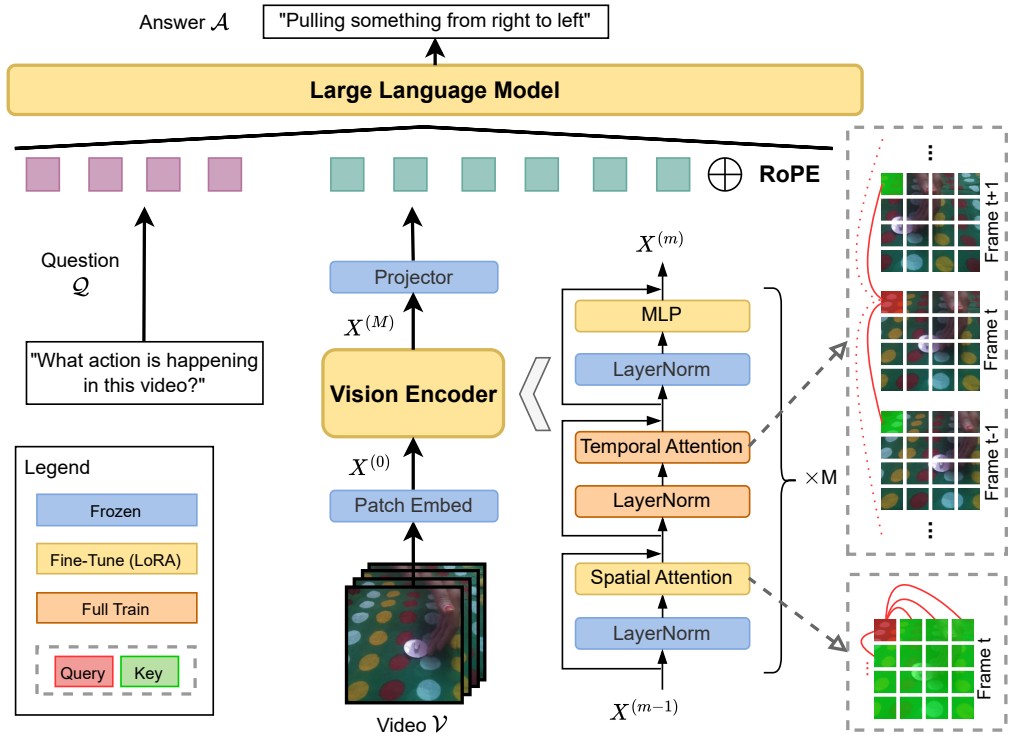

Figure 3: Our proposed STAVEQ2 architecture. Video frames are processed through transformer blocks with spatial and stacked temporal attention modules, capturing intra-frame and inter-frame dynamics. The resulting visual tokens are fed into the LLM for answer generation.

benchmarks. Therefore, a more flexible Video-LLM must be developed to effectively process both temporal and spatial information.

With STAVEQ2, we propose enhancing the Qwen2-VL Video-LLM by incorporating temporal attention blocks. As illustrated in Figure 3, STAVEQ2 builds upon the standard vision encoder architecture, by adding dedicated temporal attention mechanisms after spatial attention blocks. This design generates token embeddings enriched with spatiotemporal information, which are aligned with the LLM through a projector module for autoregressive answer generation in VQA tasks.

### 4.3 Stacked Temporal Attention in Vision Encoders

Each transformer block $m$, processes patch embeddings $X^{(m-1)} \in \mathbb{R}^{T \times N \times D}$, where $X_{t,i}^{(m-1)} \in \mathbb{R}^D$ represents the embedding of the $i$-th patch ($i = 1, \ldots, N$) in frame $t$ ($t = 1, \ldots, T$) of the input video $\mathcal{V}$. The block consists of spatial self-attention, injected temporal self-attention, and a feed-forward multilayer perceptron (MLP), with residual connections and layer normalization (LN) applied at each stage. For clarity, we describe the formulation for a single attention head. For spatial attention, queries, keys, and values are computed for each frame $t$ as:

$$Q_t^{(m)}, K_t^{(m)}, V_t^{(m)} = \text{LN}\left(X_t^{(m-1)}\right) W_Q^{(m)}, \text{LN}\left(X_t^{(m-1)}\right) W_K^{(m)}, \text{LN}\left(X_t^{(m-1)}\right) W_V^{(m)}, \quad (1)$$

where $X_t^{(m-1)} \in \mathbb{R}^{N \times D}$ is the embedding matrix for frame $t$, and $W_Q^{(m)}, W_K^{(m)}, W_V^{(m)} \in \mathbb{R}^{D \times d_s}$ are learnable projection matrices, with $d_s$ as the dimension of queries and keys. The spatial attention weights are:

$$A_t^{(m)} = \text{softmax}\left(\frac{Q_t^{(m)} K_t^{(m)T}}{\sqrt{d_s}}\right) \quad (2)$$

and the output is:

$$S_t^{(m)} = A_t^{(m)} V_t^{(m)} + X_t^{(m-1)}. \quad (3)$$

The spatial attention outputs are concatenated along the temporal dimension as $S^{(m)} = [S_1^{(m)}; \ldots; S_T^{(m)}] \in \mathbb{R}^{T \times N \times D}$.

For temporal attention, we process the spatial features for each patch $i$ across all frames, defined as $Y_i^{(m)} = [S_{1,i}^{(m)}, \ldots, S_{T,i}^{(m)}]^\top \in \mathbb{R}^{T \times D}$. The temporal attention mechanism computes:

$$Q_i'^{(m)}, K_i'^{(m)}, V_i'^{(m)} = \text{LN}\left(Y_i^{(m)}\right) W_Q'^{(m)}, \text{LN}\left(Y_i^{(m)}\right) W_K'^{(m)}, \text{LN}\left(Y_i^{(m)}\right) W_V'^{(m)}, \quad (4)$$

where $W_Q'^{(m)}, W_K'^{(m)}, W_V'^{(m)} \in \mathbb{R}^{D \times d_t}$ are learnable projection matrices, and $d_t$ is the dimension of queries and keys. The temporal attention weights are:

$$A_i'^{(m)} = \text{softmax}\left(\frac{Q_i'^{(m)} K_i'^{(m)T}}{\sqrt{d_t}}\right) \quad (5)$$

and the output is:

$$Z_i^{(m)} = A_i'^{(m)} V_i'^{(m)} + Y_i^{(m)}. \quad (6)$$

The temporal attention outputs are concatenated along the spatial dimension as $Z^{(m)} = [Z_1^{(m)}, \ldots, Z_N^{(m)}]^\top \in \mathbb{R}^{T \times N \times D}$. The block concludes with an MLP and residual connection:

$$X^{(m)} = \text{MLP}\left(\text{LN}\left(Z^{(m)}\right)\right) + Z^{(m)}, \quad (7)$$

The feature representation $X^{(M)}$ of the video $\mathcal{V}$ output from the last transformer block $M$ is projected to the LLM's embedding space, concatenated with the language embeddings of the question $\mathcal{Q}$, and passed to the LLM. This setup allows the LLM to reason jointly over visual and linguistic modalities while benefiting from temporally-aware visual embeddings.

$$\mathcal{A} = \text{LLM}\left(\text{Projector}(X^{(M)}), \ \mathcal{Q}\right) \quad (8)$$

Our key innovation is the parameter-efficient temporal attention module. By using up to four times fewer attention heads than spatial attention while maintaining the head dimension, we significantly reduce parameters (See Appendix C for the ablation studies). We apply 1D rotary position embeddings (RoPE) in the temporal attention block to encode temporal structure, unlike the 2D RoPE used for spatial position encoding in models like Qwen2-VL. This lightweight design enhances temporal modeling with minimal computational overhead, improving the quality of video features for temporally challenging VQA tasks.

## 5 Evaluation

We evaluate STAVEQ2 on several video understanding benchmarks, demonstrating that STAVEQ2 outperforms recent state-of-the-art Video-LLMs. We compare STAVEQ2 with the Qwen2-VL 2B/7B/72B and Qwen2.5-VL 7B/72B [3], as well as other state-of-the-art models. Furthermore, by applying our proposed stacked temporal attention to the InternVideo2 video foundation model, we achieve a new state-of-the-art result on the SSv2 action recognition dataset. We also conduct experiments on a variation of the SSv2 dataset and try to see how well models can learn to match similarities between videos and to see how STAVEQ2 can compare. The experiments are conducted using 64 NVIDIA A100 GPUs.

### 5.1 Training STAVEQ2

To train our STAVEQ2 model, we use a two-stage process that integrates the injected temporal attention blocks with minimal disruption to the pre-trained Qwen2-VL. We train on VQA datasets using cross-entropy loss to optimize the generated textual answers for temporally challenging tasks. To train STAVEQ2 while preserving the instruction-following capabilities of base instruction-tuned models, we curate a subset of WebVid [4], a large-scale video-caption dataset, and generate multi-turn question-answer pairs by prompting the Qwen2 7B [51] LLM. We refer to this dataset as WebVid-QA.

Table 2: Performance for fine-tuning on SSv2 (full dataset). Our InternVideo2 1B + Stacked Temporal Attention (STA) sets a new state-of-the-art on the full SSv2 dataset, outperforming the larger InternVideo2 6B model by 0.5%.

| Model | Accuracy (%) |
|---|---|
| VideoMAE V2-H [43] | 76.8 |
| VideoMAE V2-g [43] | 77.0 |
| MVD-L [45] | 76.7 |
| MVD-H [45] | 77.3 |
| InternVideo2 1B [47] | 77.1 |
| InternVideo [46] | 77.2 |
| InternVideo2 6B [47] | 77.5 |
| InternVideo2 1B + STA | **78.0**  (↑ 0.5%) |

In the first stage, we initialize the output projection layer of the temporal multi-head attention blocks to zero, ensuring the vision encoder initially behaves like the original model and preserves its pre-trained spatial modeling capabilities. We freeze all model parameters except the temporal attention blocks and the associated layer normalizations. These are trained with a linear warmup over the first steps, gradually integrating temporal attention into the encoder.

In the second stage, we introduce LoRA adapters [19] to the linear layers of both the vision encoder (attention projections and MLP) and the LLM, using a small rank to maintain parameter efficiency. The temporal attention blocks and LoRA adapters are jointly trained to align the enhanced spatiotemporal features with the LLM's linguistic reasoning. This stage ensures that the model effectively processes temporal dynamics for VQA tasks, enhancing feature quality for the LLM, as validated by its performance on temporally challenging questions.

## 5.2 Selected Benchmarks

We evaluate our method on multiple tasks and benchmarks, including full SSv2 action recognition, SSv2-VSM dataset for visual similarity matching, and three diverse video understanding benchmarks, each testing different aspects of video comprehension: **VITATECS** [29]: A diagnostic dataset disentangled from static information for temporal concept understanding of Video-LLMs across six aspects: compositionality, direction, intensity, localization, sequence, and type. **MVBench** [28]: A multimodal video understanding benchmark covering 20 challenging video tasks that cannot be effectively solved with a single frame. **Video-MME** [15]: A comprehensive benchmark for evaluating MLLMs on video understanding across 6 primary visual domains and various durations.

## 5.3 Temporal Performance Improvements

To analyze the temporal modeling capabilities of stacked temporal attention module, we enhanced the InternVideo2 vision-only model with STA and fine-tuned it on the full SSv2 dataset for action recognition. This experiment showcases STA's ability to boost temporal understanding, even in models with pre-existing joint spatiotemporal attention. The results, detailed in Table 2, reveal that InternVideo2 1B with STA achieves a new state-of-the-art accuracy of 78.0%, outperforming the larger InternVideo2 6B model by 0.5%, despite InternVideo2 1B + STA having only about 1.3B parameters. This leap in performance underscores the efficiency of STA's dedicated temporal focus, proving that the gains arise from enhanced temporal modeling rather than just a modest parameter increase. Comparisons with other models are also presented in Table 2.

## 5.4 Visual Similarity Understanding

To further investigate the temporal understanding of Video-LLMs, we also examine their ability to perform visual similarity matching. Since transformer-based models operate using self-attention, detecting similarity between tokens is a relatively straightforward task for them. Therefore, if the vision encoder is effectively capturing and representing video content, the model should be able to

Table 3: Evaluation results for fine-tuning on SSv2-VSM. The task is to decide which of two videos is similar to a third video.

| Model | Accuracy (%) |
|---|---|
| Qwen2-VL 2B | 68.65 |
| STAVEQ2 2B | **72.19** (↑ 3.5%) |
| Qwen2-VL 7B | 73.15 |
| STAVEQ2 7B | **76.05** (↑ 2.9%) |

Table 4: Accuracy (%) on video understanding benchmarks for our STAVEQ2 compared to other models. For VITATECS, aspect-wise results are shown; other benchmarks report overall accuracy. *(Video-MME without/with subtitles). † Results collected from the Video-MME leaderboard. – indicates results not reported in the original paper and unavailable from other sources.

| Model | VITATECS | | | | | | MVBench | *VMME (wo/w) |
|---|---|---|---|---|---|---|---|---|
| | Comp. | Dir. | Int. | Loc. | Seq. | Type | | |
| Qwen2-VL 2B | 80.8 | 82.1 | 69.6 | 76.1 | 72.2 | 85.9 | 63.2 | 55.6 / 60.4 |
| STAVEQ2 2B (Ours) | **81.3** | **83.0** | **70.1** | **76.9** | **72.9** | **86.6** | **65.1** | **56.2 / 61.3** |
| ST-LLM 7B [32] | – | – | – | – | – | – | 54.9 | – |
| TG-Vid 7B [21] | – | – | – | – | – | – | 56.4 | – |
| LLaVA-OneVision 7B [9] | – | – | – | – | – | – | 56.7 | 58.2 / – |
| Qwen2-VL 7B | 88.9 | 86.6 | 78.2 | 80.6 | 82.8 | 88.8 | 67.0 | 63.3 / 69.0 |
| Qwen2.5-VL 7B | 86.1 | 80.0 | 73.0 | 77.3 | 78.8 | 88.2 | 69.6 | 65.1 / 71.6 |
| STAVEQ2 7B (Ours) | **89.8** | **87.6** | **78.7** | **80.9** | **83.9** | **88.9** | **70.1** | **66.8 / 71.8** |
| LLaVA-OneVision 72B [9] | – | – | – | – | – | – | 59.4 | 66.2 / 69.5 |
| VideoLLaMA2 72B [10] | – | – | – | – | – | – | 62.0 | 61.4 / 63.1 |
| LLaVA-Video 72B† [55] | – | – | – | – | – | – | – | 70.6 / 76.9 |
| Qwen2-VL 72B | 89.8 | 87.8 | 77.9 | 85.3 | 84.8 | 90.4 | 73.6 | 71.2 / 77.8 |
| Qwen2.5-VL 72B | 92.1 | 88.9 | 81.9 | 87.1 | 89.4 | 91.8 | 70.4 | 73.3 / 79.1 |
| STAVEQ2 72B (Ours) | **92.8** | **90.1** | **82.3** | **87.9** | **90.3** | **92.8** | **74.5** | **73.9 / 79.9** |
| GPT-4o† | – | – | – | – | – | – | – | 71.9 / 77.2 |

solve visual matching problems quite well. This makes visual similarity tasks a good way to check whether the model is learning useful video representations.

Based on this idea, we create the **SSv2-VSM** (Visual-Similarity-Matching) subset, a variation of SSv2-T10 designed to evaluate whether Video-LLMs can recognize visual similarity between actions. The dataset contains 8,471 samples, each consisting of two reference videos (with different actions) and a third query video. The task is to determine whether the action in the query video matches the first, the second, or neither video. See Appendix A for prompt examples and Appendix E for further details on the subset's composition.

We evaluate Qwen2-VL 2B and 7B, along with their STAVEQ2-fine-tuned versions, on SSv2-VSM. Because this task emphasizes visual comparison rather than open-ended text generation (like action recognition), it allows us to isolate the quality of the model's video representations. As shown in Table 3, our STAVEQ2 variants outperform the base models by $2.9\%$ and $3.54\%$, respectively. This suggests that temporal features can improve performance even in tasks focused on visual similarity.

## 5.5 STAVEQ2 Evaluation on Benchmarks

We evaluate the performance of STAVEQ2 on standard video understanding benchmarks to assess general video understanding capabilities in temporally challenging scenarios. Results in Table 4 show that STAVEQ2 consistently outperforms recent state-of-the-art Video-LLMs in all of the benchmarks, demonstrating the robustness and general applicability of our method.

We conducted the VITATECS benchmark results for the Qwen2-VL and Qwen2.5-VL family, as they do not provide results for this benchmark themselves. In the VITATECS benchmark, we observe that the STAVEQ2 models outperform both their Qwen2-VL counterparts and the newer Qwen2.5-VL models, achieving the highest scores in all aspects, including direction and sequence understanding. On the Video-MME dataset, we outperform other models, including GPT-4o by 2/2.7 and LLaVA-Video 72B by 3.3/3 regarding accuracy. Our STAVEQ2 72B also outperforms other models on MVBench. Notably, STAVEQ2 7B surpasses ST-LLM 7B, a model which attempts to delegate the task of modeling spatiotemporal sequences to the LLM, by 15.2 on MVBench. Additionally, STAVEQ2 7B outperforms TG-Vid 7B by 13.7.

## 6    Conclusions

We introduced STAVEQ2, our Video-LLM architecture with stacked temporal attention modules in the vision encoder, enabling more accurate modeling of temporal relationships and action progression across video frames. Our analysis showed that existing models struggle with temporal reasoning. By enhancing temporal modeling directly at the visual encoding stage, our model improves video question answering performance. These findings suggest that temporal attention at the encoder level is crucial for better generalization and temporal understanding in Video-LLMs.

**Limitations and Future Work**    Due to resource constraints, we did not pretrain the model or train from scratch, and our experiments were limited to models up to 72B parameters. Future work can explore full pretraining and scaling beyond 72B to further evaluate the architecture's potential.

## Acknowledgments and Disclosure of Funding

This work was partially funded by the Federal Ministry for Transport (BMV), Germany ("MoToRes", 01F2271A) and by the Federal Ministry for Economic Affairs and Energy (BMWE), Germany ("ATTENTION!", 01MJ22012D).

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

# A  Prompt Examples

This section provides example prompts used for our experiments.

---

**Box 1: Example prompt with in-context examples for SSv2-T10 dataset**

**Instruction:** Look at the provided examples and answer the last question.

**Example 1 - \<video\>** The action happening in this video is:
Moving [something] from left to right.

**Example 2 - \<video\>** The action happening in this video is:
Moving [something] from right to left.

**Final Prompt - \<video\>** Now considering the previous examples, what action is happening in this video?

---

**Box 2: Example prompt for SSv2-VSM dataset**

**Instruction:** Look at the provided examples and identify which example is related to the final video.

**Example 1 - \<video\>** The action happening in this video is:
Moving [something] away from [something].

**Example 2 - \<video\>** The action happening in this video is:
Moving [something] closer to [something].

**Final Prompt - \<video\>** Now considering the previous examples, is there any action related to this video? If not, respond with "No related action" and if there is, respond with the example number and action.

---

**Box 3: Example Prompt for Evaluation**

**Instruction:** Look at the ground truth and the LLM's answer. Decide whether the LLM's answer matches the ground truth.

**Ground Truth:** Pulling [something] from left to right

**LLM Answer:** The action is moving something from right to left on the floor

**Question:** Based on the ground truth, is the LLM answer correct? Answer with a simple "Yes" or "No".

---

Box 1 illustrates the prompt structure used in our few-shot experiments on the SSv2-T10 dataset. In this setup, the model is presented with one or more in-context examples, each consisting of a video and its corresponding action label. After reviewing these examples, the model is tasked with inferring the action occurring in a new query video. This format encourages the model to generalize from the provided samples and demonstrate its capability to recognize similar actions.

Box 2 presents the prompt format used for the SSv2-VSM dataset, which is designed to evaluate the model's ability to perform similarity matching. The model is provided with multiple labeled video examples and a final query video, and it must determine whether any of the examples depict an action related to the one shown in the query. If a related action exists, the model is expected to return the example number along with the action label; otherwise, it should respond with "No related action." This task format emphasizes fine-grained action discrimination and serves as a valuable test of the model's capacity for visual-semantic matching.

Table 5: Action categories and their relative frequencies in SSv2-T10.

| Action | Frequency (%) |
|---|---|
| Pulling [something] from left to right | 14.68 |
| Pulling [something] from right to left | 14.97 |
| [Something] falling like a rock | 13.12 |
| Picking [something] up | 9.26 |
| Throwing [something] in the air and letting it fall | 7.31 |
| Throwing [something] in the air and catching it | 8.90 |
| Moving [something] away from [something] | 8.61 |
| Moving [something] closer to [something] | 8.55 |
| Rolling [something] on a flat surface | 11.85 |
| Poking a stack of [something] so the stack collapses | 2.74 |

Table 6: Ablation study results on order of spatial and temporal attention and head scaling in STAVEQ2 2B on SSv2-T10. Accuracy (%) is reported, with the best result in bold.

| Model | Attention Order | Head Scale | Acc (%) |
|---|---|---|---|
| Qwen2-VL 2B | – | 1.0 | 73.14 |
| STAVEQ2 2B | Spatial First | 1.0 | 58.34 |
| STAVEQ2 2B | Spatial First | 0.5 | 71.18 |
| STAVEQ2 2B | Temporal First | 0.25 | 73.20 |
| STAVEQ2 2B | Spatial First | 0.25 | **76.04** |

In open-ended generation, the model's output may not precisely match the expected action name. To address this, we employ another LLM, Qwen2-7B, as a judge to evaluate the responses generated by Video-LLMs. Box 3 provides the evaluation prompt. Given a ground truth label and the model's predicted answer, the judge determines whether the prediction is correct, returning a binary "Yes" or "No" response.

## B  SSv2-T10 Composition

Table 5 lists the action categories in the SSv2-T10 split and their relative frequencies (percentage of examples). These ten actions were used both to select representative in-context examples and to design the prompts used in our experiments.

## C  Ablation Studies on STA Architecture

To investigate the design of our stacked temporal attention, we conduct ablation studies on STAVEQ2 2B, fine-tuned on the SSv2-T10 dataset. The first study examines the internal configuration of STA-enhanced transformer blocks, including the order of spatial and temporal attentions (i.e. whether temporal attention should be placed either before or after spatial attention) and the number of temporal attention heads. The second study investigates the placement of STA-enhanced transformer blocks across the vision encoder.

As summarized in Table 6, positioning temporal attention after spatial attention with a head scaling factor of 0.25 achieves the highest accuracy (76.04%), outperforming the baseline Qwen2-VL 2B (73.14%) by 2.90%. Reducing the number of temporal attention heads (e.g., 0.25 vs. 1.0 relative to the baseline number of heads) enhances performance, likely due to improved regularization and focus on critical temporal features, particularly when there is limited data, aligning with our emphasis on parameter efficiency. Placing temporal attention before spatial attention (73.20% at 0.25 scale) yields slightly lower performance, indicating that processing spatial context first enhances temporal modeling. These results validate the design of our stacked temporal attention, especially for fine-grained temporal understanding tasks.

Table 7: Ablation study results on number and placement of STA-enhanced transformer blocks in STAVEQ2 2B on SSv2-T10. Accuracy (%) is reported, with the best result in bold.

| Model | # Temporal Blocks | Placement | Acc (%) |
|---|---|---|---|
| Qwen2-VL 2B | – | – | 73.14 |
| STAVEQ2 2B | 16 | Uniform | 74.73 |
| STAVEQ2 2B | 16 | First blocks | 74.97 |
| STAVEQ2 2B | 32 | All blocks | **76.04** |

Table 8: Performance of InternVideo2-Chat, fine-tuned on SSv2-T10. Adding stacked temporal attention (STA) leads to a significant accuracy gain.

| Method | Acc (%) |
|---|---|
| InternVideo2-Chat 8B | 84.17 |
| InternVideo2-Chat 8B + STA | **95.18** (↑ 11.01%) |

Building on the optimal internal configuration from Table 6, we assess the impact of the number and placement of STA-enhanced transformer blocks across the vision encoder in STAVEQ2. According to the results summarized in Table 7, using 32 STA-enhanced transformer blocks across all layers achieves the highest accuracy (76.04%), outperforming the baseline Qwen2-VL 2B (73.14%) by 2.90%. Reducing the number of STA-enhanced blocks to 16, whether distributed uniformly (74.73%) or concentrated in early layers (74.97%), results in a lower performance. The minimal difference between uniform and early-layer placement indicates that strategic block positioning has less impact than the total number of blocks.

## D   STA Enhancement Effect on InternVideo2-Chat

To complement the qualitative analysis in Section 3.3 (Figures 2a–2c), Table 8 provides the corresponding quantitative performance metrics for InternVideo2-Chat 8B on SSv2-T10, both before and after applying stacked temporal attention (STA). Note that InternVideo2-Chat is limited to processing up to 8 input frames, making it suitable only for short-video benchmarks like SSv2-T10 and excluding it from evaluations on longer-video datasets such as those in Table 4. The baseline InternVideo2-Chat 8B model achieves an accuracy of 84.17%. With the integration of STA, the performance improves significantly, reaching 95.18%—a substantial gain of 11.01%. These results underscore the effectiveness of STA in boosting the temporal understanding capabilities of Video-LLMs, particularly for fine-grained action recognition tasks.

## E   SSv2-VSM Dataset Composition

As described in the paper, each sample in SSv2-VSM dataset consists of two reference videos (with different actions) and a third query video. The task is to determine whether the action in the query video matches the first, the second, or neither video. For the SSv2-VSM dataset, we explored the optimal ratio of positive to negative samples for fine-tuning the models. Positive samples consist of reference videos where one matches the query video's action, while negative samples have no matching actions with the query video.

Table 9 shows that increasing the proportion of positive samples generally enhances performance, with similarity matching accuracy improving from 25.52% at 50% positive samples to a peak of 71.25% at 80% positive samples. However, accuracy drops to 49.18% at 91% positive samples, indicating that an excessively high proportion of positive samples may reduce dataset diversity and hinder generalization.

In our experiments reported in Table 9, we include textual descriptions of the actions happening in each reference video. However, removing these textual descriptions increases task difficulty, as the

Table 9: Context-Selection fine-tuning results with varying positive-negative sample ratios. Similarity Matching evaluates whether the model can correctly identify a relevant context sample among distractors when such a sample is present.

| Dataset Composition | Accuracy (%) |
|---|---|
| 50% positive | 25.52 |
| 80% positive | **71.25** |
| 91% positive | 49.18 |

similarity matching accuracy at 80% positive samples drops from 71.25% to 68.65% by removing the descriptions. Consequently, to evaluate the models on a more challenging task, we excluded textual descriptions for reference videos in the main experiments reported in the paper.

# F   Prompting Experiments

Recent work [39] systematically investigated the factors affecting multi-modal in-context learning performance in image and text modalities, demonstrating that instruction placement and modality ordering can impact performance. To understand if these findings generalize to the video domain, we conducted a parallel set of controlled experiments on the SSv2-VSM dataset.

First, we evaluated the impact of instruction placement by testing four prompting strategies, adapting the methodology from [39]: **No-Instruction**, where the prompt contains no explicit task instruction; **Introductive**, where a task description appears once at the start of the prompt, before the context examples; **Introductive-Summative**, where an introductory instruction is used, along with a summary instruction that appears after the context examples but before the final query; and **Intra-demonstration**, where the task instruction is repeated within each demonstration example. As shown in Table 10, the choice of instruction style had a minimal effect on the performance of most models. However, we observed a notable exception: InternVideo2-Chat's performance improved significantly (by 9%) with the **Introductive-Summative** strategy. Sample prompts illustrating each strategy are provided in Box 4–7.

Second, we tested the effect of intra-demonstration modality ordering—that is, whether the sequence of the video and its corresponding text tag within each context sample affects model understanding. We compared two formats: **Text-Video** (where text precedes the <video> tag) and **Video-Text** (where the <video> tag appears first). The results in Table 11 again show a strong model-dependent sensitivity. While the ordering had minimal impact on the performance of most models, it was a critical factor for InternVideo2-Chat. This model performed significantly better (37.06% vs. 15.98%) when the video was presented first (**Video-Text**), aligning with findings for image-based models in [39].

---

**Box 4: No-Instruction example prompt for SSv2-VSM**

**Instruction:**

**Example 1 - <video>** The action happening in this video is: Throwing [something] in the air and catching it.

**Example 2 - <video>** The action happening in this video is: Pulling [something] from left to right.

**Final Prompt - <video>** now considering the previous examples, is there any action related to this video? If not, respond with `No related action` and if there is, respond with the example number and action.

---

**Box 5: Introductive example prompt for SSv2-VSM**

**Instruction:** Look at the provided videos and identify which video is related to the final video.

**Example 1 - \<video\>** The action happening in this video is: Throwing [something] in the air and catching it.

**Example 2 - \<video\>** The action happening in this video is: Pulling [something] from left to right.

**Final Prompt - \<video\>** now considering the previous examples, is there any action related to this video? If not, respond with `No related action` and if there is, respond with the example number and action.

---

**Box 6: Introductive-Summative example prompt for SSv2-VSM**

**Instruction:** Look at the provided videos and identify which video is related to the final video.

**Example 1 - \<video\>** The action happening in this video is: Throwing [something] in the air and catching it.

**Example 2 - \<video\>** The action happening in this video is: Pulling [something] from left to right.

**Summary:** In summary, the two given videos each contain an action taking place, which has been provided to you. You need to recognize these actions and keep them in mind for the next question.

**Final Prompt - \<video\>** now considering the previous examples, is there any action related to this video? If not, respond with `No related action` and if there is, respond with the example number and action.

---

**Box 7: Intra-demonstration example prompt for SSv2-VSM**

**Instruction:**

**Example 1 - \<video\>** The action happening in this video is: Throwing [something] in the air and catching it. So this is the video number 1, remember the video and the action taking place, you need to see if another video has a similar action later.

**Example 2 - \<video\>** The action happening in this video is: Pulling [something] from left to right. So this is the video number 2, remember the video and the action taking place, you need to see if another video has a similar action later.

**Final Prompt - \<video\>** now considering the previous examples, is there any action related to this video? If not, respond with `No related action` and if there is, respond with the example number and action.

## G   Attention Visualization

To qualitatively evaluate the impact of stacked temporal attention (STA), we visualized the attention maps generated by InternVideo2 1B vision only model before and after applying STA. Figure 4 shows a person poking a lighter so that it falls, labeled with the class *poking [something] so that it falls over*. Before applying STA, the attention maps show that the model places minimal focus on the lighter, despite the fact that the action class is primarily defined by the lighter's movement and fall. After

Table 10: Prompt structure impact on SSv2-VSM performance. We compare four prompting styles: No-Instruction, Introductive (task instruction at the beginning), Introductive-Summative (instructions both at the beginning and after context), and Intra-demonstration (instruction repeated before each context sample).

| Model | No-Instruction | Introductive | Introductive-Summative | Intra-demonstration |
|---|---|---|---|---|
| LLaVA-NeXT-Video 7B | 20.08% | 20.11% | 20.11% | 20.11% |
| Qwen2-VL 2B | 22.76% | 21.29% | 21.46% | 24.97% |
| Qwen2-VL 7B | 20.14% | 20.11% | 20.22% | 20.28% |
| InternVideo2-Chat 8B | 35.94% | 37.06% | 45.02% | 34.94% |

Table 11: Effect of intra-context tag ordering on SSv2-VSM performance. We compare two context formats: Text-Video, where the text appears before the video tag in each context item, and Video-Text, where the video tag comes first.

| Model | Text-Video | Video-Text |
|---|---|---|
| LLaVA-NeXT-Video 7B | 20.11% | 20.11% |
| Qwen2-VL 2B | 24.09% | 21.29% |
| Qwen2-VL 7B | 22.32% | 20.11% |
| InternVideo2-Chat 8B | 15.98% | 37.06% |

incorporating STA, the model's attention shifts significantly toward the lighter, especially as it begins to fall, indicating improved temporal modeling and relevance attribution.

These visualizations underscore the effectiveness of stacked temporal attention in helping the model focus on temporally relevant regions of the video. Especially in cases where the action depends on subtle object movements over time, STA enhances the model's ability to localize and interpret key interactions, thereby improving classification performance.

## H   Extra Models

While our primary contributions center on enhancing the Qwen2-VL architecture with stacked temporal attention in STAVEQ2, we further validate the broad applicability of this mechanism by extending it to several very recent and competitive Video-LLM architectures. These extensions demonstrate that STA is not tied to a specific base model but provides consistent improvements in temporal understanding across diverse design paradigms—including those that already incorporate advanced temporal modeling techniques. Due to resource constraints, we focused on representative recent models that align closely with our evaluation benchmarks.

For the newer Qwen2.5-VL, we introduce STAVEQ2.5, which integrates dedicated temporal attention blocks into its vision encoder. Qwen2.5-VL employs windowed attention in most layers and full spatial self-attention in only four layers; however, our approach remains unaffected, as each patch in the STA block attends to corresponding patches across all frames, bypassing the limitations imposed by windowed attention. This design preserves the temporal attention mechanism of STAVEQ2, ensuring consistent temporal modeling. We train STAVEQ2.5 using the same two-stage strategy employed for STAVEQ2, leveraging the WebVid-QA dataset.

We also apply STA to VideoRoPE 7B [50], a state-of-the-art model based on Qwen2-VL that enhances temporal awareness through improved positional embeddings for output feature tokens fed to the LLM. In contrast, our STA targets token-level temporal interactions directly within the vision encoder, making the two approaches complementary rather than competing. Our experiments confirm this synergy: integrating STA with VideoRoPE yields further gains, highlighting how dedicated temporal

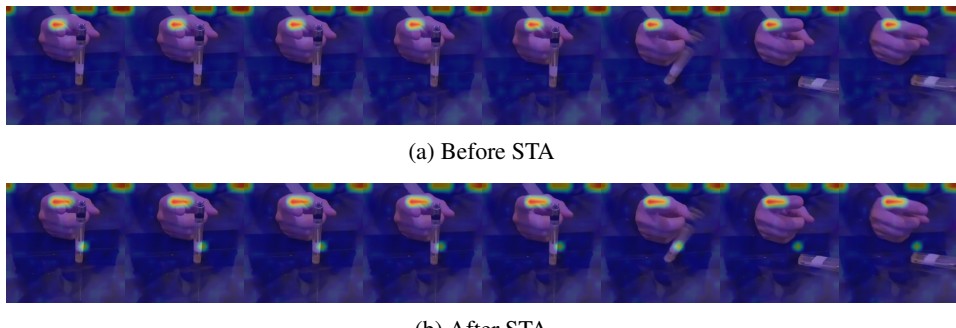

(a) Before STA

(b) After STA

Figure 4: Attention maps for the action *poking [something] so that it falls over*.

Table 12: Accuracy (%) on video understanding benchmarks for our STAVEQ2.5 compared to other models. For VITATECS, aspect-wise results are shown; other benchmarks report overall accuracy. IV2.5-Chat refers to InternVideo2.5-Chat. *(Video-MME without/with subtitles). † Results collected from the Video-MME leaderboard. – indicates results not reported in the original paper and unavailable from other sources.

| Model | VITATECS | | | | | | MVBench | *VMME (wo/w) |
|---|---|---|---|---|---|---|---|---|
| | Comp. | Dir. | Int. | Loc. | Seq. | Type | | |
| Qwen2-VL 2B | 80.8 | 82.1 | 69.6 | 76.1 | 72.2 | 85.9 | 63.2 | 55.6 / 60.4 |
| STAVEQ2 2B (Ours) | **81.3** | **83.0** | **70.1** | **76.9** | **72.9** | **86.6** | **65.1** | **56.2 / 61.3** |
| ST-LLM 7B | – | – | – | – | – | – | 54.9 | – |
| TG-Vid 7B | – | – | – | – | – | – | 56.4 | – |
| LLaVA-OneVision 7B | – | – | – | – | – | – | 56.7 | 58.2 / – |
| VideoRoPE | 81.1 | 81.8 | 60.9 | 79.4 | 80.7 | 85.8 | 57.3 | 61.6 / – |
| VideoRoPE + STA (Ours) | 81.9 | 82.9 | 61.8 | 79.9 | 81.3 | 86.3 | 59.2 | 62.5 / – |
| Qwen2-VL 7B | 88.9 | 86.6 | 78.2 | 80.6 | 82.8 | 88.8 | 67.0 | 63.3 / 69.0 |
| Qwen2.5-VL 7B | 86.1 | 80.0 | 73.0 | 77.3 | 78.8 | 88.2 | 69.6 | 65.1 / 71.6 |
| STAVEQ2 7B (Ours) | 89.8 | 87.6 | 78.7 | 80.9 | 83.9 | 88.9 | 70.1 | **66.8** / 71.8 |
| STAVEQ2.5 7B (Ours) | 88.0 | 82.1 | 74.2 | 77.9 | 79.7 | 88.9 | 70.3 | 66.2 / **72.5** |
| InternVideo2.5-Chat 8B | 91.3 | 88.7 | 82.0 | 84.8 | 84.7 | 91.0 | 75.7 | 65.1 / – |
| IV2.5-Chat 8B + STA (Ours) | **91.6** | **89.7** | **82.7** | **85.6** | **85.8** | **91.3** | **76.8** | 65.9 / – |
| LLaVA-OneVision 72B | – | – | – | – | – | – | 59.4 | 66.2 / 69.5 |
| VideoLLaMA2 72B | – | – | – | – | – | – | 62.0 | 61.4 / 63.1 |
| LLaVA-Video 72B | – | – | – | – | – | – | – | 70.6 / 76.9 |
| Qwen2-VL 72B | 89.8 | 87.8 | 77.9 | 85.3 | 84.8 | 90.4 | 73.6 | 71.2 / 77.8 |
| Qwen2.5-VL 72B | 92.1 | 88.9 | 81.9 | 87.1 | 89.4 | 91.8 | 70.4 | 73.3 / 79.1 |
| STAVEQ2 72B (Ours) | 92.8 | 90.1 | **82.3** | 87.9 | 90.3 | 92.8 | **74.5** | 73.9 / **79.9** |
| STAVEQ2.5 72B (Ours) | **93.1** | **90.9** | 82.1 | **88.0** | **90.8** | **93.3** | 72.4 | **74.2** / 79.8 |
| GPT-4o† | – | – | – | – | – | – | – | 71.9 / 77.2 |

attention can amplify existing enhancements. Finally, we extend STA to InternVideo2.5-Chat 8B [48], a recent evolution of the InternVideo family that supports longer video inputs (unlike InternVideo2-Chat that is limited to processing 8 input frames). InternVideo2.5-Chat is trained using the same approach as other models and evaluated on the full set of benchmarks.

Evaluation on video understanding benchmarks—VITATECS, MVBench, and Video-MME—demonstrates the effectiveness of these extensions in handling temporally complex scenarios. The results show that our STA approach not only enhances Qwen2-VL but also improves these newer architectures, confirming its wide applicability.

As shown in Table 12, STAVEQ2.5 consistently outperforms its base model, Qwen2.5-VL, across VITATECS, MVBench, and Video-MME, validating the effectiveness of our stacked temporal attention approach in enhancing video understanding. Notably, STAVEQ2.5 72B surpasses STAVEQ2 72B on most benchmarks, demonstrating superior performance of our approach when applied to the

newer Qwen2.5-VL architecture. Similarly, applying STA to VideoRoPE and InternVideo2.5-Chat yields consistent gains, further highlighting the method's robustness and its ability to synergize with orthogonal temporal enhancements. These results underscore the adaptability of our stacked temporal attention mechanism, which drives performance improvements across Video-LLMs of varying scales and architectures, excelling in tasks that demand sophisticated temporal understanding.

