# OpenReview forum: "Enhancing Temporal Understanding in Video-LLMs through Stacked Temporal Attention in Vision Encoders"
_NeurIPS.cc/2025/Conference — NeurIPS 2025 poster_

### Official Review · Reviewer_tYNw · 2025-06-27

**Clarity:** 2
**Significance:** 2
**Originality:** 3
**Rating:** 3
**Confidence:** 3

**Summary:**

The paper presents STAVEQ2, a novel Video-LLM architecture that incorporates stacked temporal attention modules within the vision encoder to enhance temporal understanding in video question answering tasks. The authors identify critical limitations in current Video-LLMs in capturing fine-grained temporal dynamics and propose an improved model that addresses these shortcomings. The paper demonstrates significant performance improvements on several benchmarks, including VITATECS, MVBench, and Video-MME, with gains of up to +5.5%. Ablations confirm STA’s efficacy, and application to video-only models (e.g., InternVideo2) yields new SOTA on SSv2 action recognition (78.0%).

**Questions:**

1. How many frames were used per video? Were frames uniformly sampled or dynamically selected?
2. The authors should explicitly address SSv2-T10 dataset statistics?
3. Can the author explain the number of transformer blocks in the experiment? And add the ablation experiment of number of transformer blocks.
4. Can the author add comparisons to VTG-LLM, InternVideo2.5, and VideoPrism?
5. Should the authors provide clearer motivation, focus their problem statement more precisely, and adequately address the other concerns raised, I would consider raising my score accordingly.

**Ethical Concerns:**

["NO or VERY MINOR ethics concerns only"]

**Final Justification:**

I will maintain my original score. While I appreciate the clarifications provided, several critical concerns remain unaddressed.

Regarding W1 & Q1, the response explicitly details frame sampling only for Qwen2-VL and InternVideo2, but fails to specify the frame sampling strategy or counts for other baseline models evaluated in the main paper.

Concerning W2 & Q2, the rebuttal states 13,984 samples in SSv2-T10 dataset.  However, this contradicts the main paper's assertion of 14,462 samples (Section 3.1).

For W4 & Q4 and Table 2, I was unable to reproduce the reported results for MVBench and VideoMME on InternVideo2.5. To validate these claims, please provide the exact experimental configuration: (1) the specific codebase/library versions used (e.g., PyTorch, Transformers, lmms-eval, VLMEvalKit), (2) config files, sampling strategies, input resolution, and batch size. Without this granularity, the reproducibility and validity of Table 2’s results remain questionable.

**Limitations:**

yes

**Paper Formatting Concerns:**

No Paper Formatting Concerns

**Quality:**

3

**Strengths And Weaknesses:**

Strengths
1. The introduction of stacked temporal attention modules directly within the vision encoder is a novel approach that effectively addresses the limitations of current Video-LLMs in temporal understanding. This design explicitly equips the vision encoder with temporal attention, enabling better capture of action progression and frame relationships.
2. The paper provides extensive experiments across multiple benchmarks, demonstrating the robustness and general applicability of the proposed method. The results show significant improvements over recent state-of-the-art Video-LLMs, highlighting the effectiveness of the temporal attention modules.
3. The authors conduct a thorough analysis of the limitations of current Video-LLMs, providing valuable insights into their shortcomings in temporal reasoning. This analysis motivates the proposed approach and underscores the importance of temporal modeling in video understanding tasks.
4. The proposed temporal attention modules are parameter-efficient, using up to four times fewer attention heads than spatial attention while maintaining head dimension. This design minimizes computational overhead while significantly enhancing temporal modeling.

Weaknesses.
1. All experiments omit the number of frames per video used for evaluation, hindering reproducibility and fair comparison.  This must be clarified.
2. The SSv2-T10 dataset’s class distribution, train/test splits  are undefined (Sec 3.1), and Appendix fails to clarify the list of selected action classes.
3. While Appx.B tests STA placement/head scaling, critical ablation of Impact of STA depth (number of transformer blocks) is missing.
4. Recent Video-LLMs (e.g., VTG-LLM[1], Internvideo2.5[2],InternVL3[3]) and temporal modeling methods (e.g., VideoPrism[4]) are absent from comparisons.  This omission undermines claims of architectural superiority.
5. The Introduction lacks sufficient clarity. The challenges presented are too vague. Temporal understanding is a broad-ranging task; the paper should focus on addressing a specific aspect of it. Proposing a method to solve temporal understanding in its entirety is impractical. The motivation needs to be significantly more concrete.

[1] Guo Y, Liu J, Li M, et al. Vtg-llm: Integrating timestamp knowledge into video llms for enhanced video temporal grounding[C]//Proceedings of the AAAI Conference on Artificial Intelligence. 2025, 39(3): 3302-3310.

[2] Wang Y, Li X, Yan Z, et al. InternVideo2. 5: Empowering Video MLLMs with Long and Rich Context Modeling[J]. arXiv preprint arXiv:2501.12386, 2025.

[3] Zhu J, Wang W, Chen Z, et al. Internvl3: Exploring advanced training and test-time recipes for open-source multimodal models[J]. arXiv preprint arXiv:2504.10479, 2025.

[4] Zhao L, Gundavarapu N B, Yuan L, et al. Videoprism: A foundational visual encoder for video understanding[J]. arXiv preprint arXiv:2402.13217, 2024.

---

> ### Author Rebuttal · Authors · 2025-07-31
>
> We thank you for the thoughtful and detailed comments. We address each concern below and provide clarifications, additional results, and plans for the camera-ready version.
>
> **W1 & Q1:** Number of frames per video clip
>
> Thank you for pointing this out. For most experiments, we sample frames at **a fixed rate of 2 fps** across the entire video, which is the default for the Qwen2-VL architecture. This dynamic approach ensures the number of frames scales with the video's duration. For instance, a standard 10-second clip from MVBench yields 20 frames, while shorter SSv2 videos (typically 2-6 seconds) result in 4-12 frames. For InternVideo2, we sample exactly 8 frames per video to comply with its fixed 8-frame input requirement. We will ensure these details are explicitly stated in the camera-ready version.
>
> **W2 & Q2:** SSv2-T10 dataset statistics
>
> Thanks for highlighting this point. For zero-shot experiments, we used a total of **3392 samples**, and for training on the **SSv2-T10** subset, we had **8476 training** and **2116 testing** samples. The distribution of the classes in all the experiments is as follows:
>
>  - Pulling [something] from left to right – 14.68%
>
> - Pulling [something] from right to left – 14.97%
>
> - [Something] falling like a rock – 13.12%
>
> - Picking [something] up – 9.26%
>
> - Throwing [something] in the air and letting it fall – 7.31%
>
> - Throwing [something] in the air and catching it – 8.90%
>
> - Moving [something] away from [something] – 8.61%
>
> - Moving [something] closer to [something] – 8.55%
>
> - Rolling [something] on a flat surface – 11.85%
>
> - Poking a stack of [something] so the stack collapses – 2.74%
>
> We will add the full class list, distribution, and split details in the Appendix.
>
>
> **W3 & Q3:** Missing ablation on STA depth
>
> We appreciate this suggestion. Since the computational complexity of temporal attention is lower than that of spatial attention—because the number of frames is smaller than the number of patches in each frame—we found no compelling reason to limit these layers to only certain transformer blocks. However, they could conceptually be placed adaptively across the transformer blocks in the vision encoder.
>
> To assess the impact of STA depth, we conducted two additional ablation studies, in addition to those in Appendix B, that varied the number and placement of temporal blocks within the transformer blocks. The results are summarized in Table 1 of our rebuttal, illustrating how different configurations affect the model’s accuracy. As shown, reducing the number of temporal blocks from 32 to 16 seems to decrease performance. Additionally, the placement of 16 temporal blocks—whether distributed uniformly or concentrated in the early layers of the vision encoder—does not result in significant differences in performance.  We will add these experiments to Appendix B, Table 1, in our camera-ready version.
>
> Table 1
>
> | Model        | Number of Temporal Blocks | Temporal Blocks Placement | Accuracy (%) |
> | :----------: | :-----------------------: | :-----------------------: | :----------: |
> | STAVEQ2 2B| 32 | All Blocks | 76.04 |
> | STAVEQ2 2B| 16                        | Uniform                   | 74.73        |
> | STAVEQ2 2B| 16                        | First Blocks              | 74.97        |
>
>
> **W4 & Q4:** Need for more baselines
>
> Thank you for this crucial point regarding comparisons to recent state-of-the-art models. To address this, we conducted two additional experiments for this rebuttal, using two highly competitive baseline models, **InternVideo2.5 [1]** and **VideoRoPE [2]**, and applied our STAVE method to them. These models represent distinct and important lines of work aimed at improving temporal understanding. InternVideo2.5 utilizes architectural innovations like hierarchical token merging, while VideoRoPE focuses on enhancing positional encoding. As shown in Table 2 of our rebuttal, applying STAVE consistently improves the performance of both models across the benchmarks. This result suggests that our approach is orthogonal and complementary; it enhances the vision encoder's core temporal processing in a way that provides additive benefits to models with different architectural strengths.
>
> Regarding the other models you mentioned, VTG-LLM[3] is highly specialized for temporal grounding, which differs significantly from our focus on general video question-answering, making direct comparison less meaningful. VideoPrism[4] is a vision-only foundation model, complicating a straightforward comparison with our full Video-LLM. Finally, InternVL3[5] was released very shortly before the submission deadline and employs test-time scaling techniques that complicate a fair comparison. While we aimed to be thorough, we must also note that our limited computational resources make it infeasible to benchmark against every recently published architecture.
>
> Table 2
>
> | Model                     | Comp. | Dir | Int. | Loc. | Seq. | Type | MVBench | VideoMME(wo) |
> | :-----------------------: | :---: | :--: | :--: | :--: | :--: | :--: | :-----: | :----------: |
> | VideoRoPE[1]              | 81.1  | 81.8 | 60.9 | 79.4 | 80.7 | 85.8 | 57.3    | 61.6         |
> | VideoRoPE+STAVE           | **81.9** | **82.9** | **61.8** | **79.9** | **81.3** | **86.3** | **59.2** | **62.5** |
> | InternVideo2.5-Chat[2]    | 91.3  | 88.7 | 82   | 84.8 | 84.7 | 91   | 75.7    | 65.1         |
> | InternVideo2.5-Chat+STAVE | **91.6** | **89.7** | **82.7** | **85.6** | **85.8** | **91.3** | **76.8** | **65.9** |
>
>
> **W5:** Introduction Clarity and Motivation
>
> Thank you for this feedback. While other reviewers noted that our problem is "well-identified" (**Reviewer q8eF**) and that we provide a "great analysis" of current limitations (**Reviewer rKct**), we want to clarify, that by temporal understanding, we are not addressing the topic “in its entirety”. Our work focuses specifically on video question answering (Video QA) tasks, particularly those that are temporally challenging. Our motivation stems from the observation that while current Video-LLMs are effective generalist models, they often struggle with questions requiring a deeper comprehension of temporal relationships and actions within the video. Our central hypothesis is that this failure is due to a lack of explicit temporal modeling mechanisms within the vision encoders of these models. To address this diagnosed weakness, we introduce STAVEQ2 and validate the hypothesis by showing significant performance gains over the baseline across all the benchmarks. We show this method's general applicability and consistent improvement over the baselines by subsequently it to other modern Video-LLMs.
>
>
> **References**
>
> [1] InternVideo2.5: Empowering Video MLLMs with Long and Rich Context Modeling, Yi Wanget. al., arXiv preprint arXiv:2501.12386 2025.
>
> [2] VideoRoPE: What Makes for Good Video Rotary Position Embedding?, Xilin Wei et. al., International Conference on Machine Learning 2025.
>
> [3] Vtg-llm: Integrating timestamp knowledge into video llms for enhanced video temporal grounding,  Guo Y, Liu J, Li M, et al., Proceedings of the AAAI Conference on Artificial Intelligence. 2025
>
> [4] Videoprism: A foundational visual encoder for video understanding, Zhao L, Gundavarapu N B, Yuan L, et al., arXiv preprint arXiv:2402.13217, 2024.
>
> [5] Internvl3: Exploring advanced training and test-time recipes for open-source multimodal models, Zhu J, Wang W, Chen Z, et al., arXiv preprint arXiv:2504.10479, 2025.

---

> > ### Comment · Reviewer_tYNw · 2025-08-05
> >
> > Thank you for the detailed rebuttal and the additional experiments. I will maintain my original score. While I appreciate the clarifications provided, several critical concerns remain unaddressed.
> >
> > Regarding W1 & Q1, the response explicitly details frame sampling only for Qwen2-VL and InternVideo2, but fails to specify the frame sampling strategy or counts for other baseline models evaluated in the main paper.
> >
> > Concerning W2 & Q2, the rebuttal states 13,984 samples in SSv2-T10 dataset.  However, this contradicts the main paper's assertion of 14,462 samples (Section 3.1).
> >
> > For W4 & Q4 and Table 2, I was unable to reproduce the reported results for MVBench and VideoMME on InternVideo2.5. To validate these claims, please provide the exact experimental configuration: (1) the specific codebase/library versions used (e.g., PyTorch, Transformers, lmms-eval, VLMEvalKit), (2) config files, sampling strategies, input resolution, and batch size. Without this granularity, the reproducibility and validity of Table 2’s results remain questionable.

---

> ### Author Response · Authors · 2025-08-03
>
> Dear Reviewer,
>
> Thank you for your initial feedback. We have submitted a detailed rebuttal addressing your comments and would greatly appreciate it if you could review our responses and share any further questions or thoughts. Your engagement is important for clarifying key points, and we’re happy to provide additional details if needed.
>
> Thank you for your time and effort!

---

> ### Author Response · Authors · 2025-08-06
>
> Thank you for your feedback during this discussion phase. Below, we provide further clarification to resolve your concerns. If any issues remain or if you have additional feedback, we encourage you to share it, and we will promptly respond to ensure all concerns are fully addressed.
>
> **Regarding W1 & Q1:**
>
> We note that in Section 5.5, Table 4 of the paper, all benchmarks for models other than our STAVEQ2 are **sourced from their original papers** or the Video-MME leaderboard, as indicated in the table caption. The only exception is the VITATECS benchmark for the Qwen2-VL and Qwen2.5-VL models, where we manually conducted the benchmark to further evaluate our model’s performance against its baseline (stated in line 286 of the paper), using a frame rate of 2fps, as reported in our rebuttal.
>
> **Regarding W2 & Q2:**
>
> In Section 3.1 of the paper, we report 14,462 samples for the SSv2-T10 dataset, derived from the parent SSv2 dataset by selecting 10 classes, comprising 10,592 samples in the training set, 1,690 in the validation set, and 2,180 in the test set of the original dataset. Your comment suggests a discrepancy by adding the 3,392 samples used in our zero-shot and in-context experiment to the 8,476+2,116 samples used in the training experiment, yielding 13,984 samples. This interpretation is incorrect, as the 3,392 samples for the zero-shot experiment are **randomly** drawn from the SSv2-T10 dataset and **overlap** with the samples used in the training experiment. These experiments are distinct, reusing samples from the same dataset, so their sample counts cannot be summed. Since these Video-LLMs had not previously encountered SSv2 dataset, this approach is valid and ensures a fair evaluation.
>
>
> - **Total** SSv2-T10 size: 10,592 + 1,690 + 2,180 = **14,462**
>
> - **Subset** size sampled for zer-shot and in-context experiments: **3,392**
>
> - **Subset** size used for training experiments (train+test): **8,476 + 2,116**
>
> To address your concern and clarify this in the camera-ready version, we can modify the training experiment to use the full 10,592 training samples and 2,180 test samples of SSv2-T10 if preferred.
>
> We should also note that for the Internvideo2+STAVE vision model (Section 5.3, Table 2), explicitly trained for the SSv2 classification task, we used the full SSv2 training set for training and the full test set for evaluation to ensure a fair comparison with other models.
>
> **Regarding W4 & Q4:**
>
> Since we **could not share code** for our **newly implemented** InternVideo2.5+STAVE experiment **during the rebuttal phase**, we kindly ask for clarification on which codebase or implementation you used to attempt reproduction.
>
> To help clarify our setup, below are the exact configurations we used:
>
> Codebase and Library Versions:
>
> - We used the official InternVideo2.5-Chat model from Hugging Face as the base.
>
> - torch==2.4.0
>
> - transformers==4.50.3
>
> - trl==0.15.1
>
> - peft==0.4.0
>
> - accelerate==0.34.2
>
> - lmms_eval==0.3.0
>
> Training Configuration:
>
> - Input: adaptive frame numbers with fps of 2
> - Temporal attention head number: 16
> - Batch size: 1
> - Gradient accumulation steps: 8
>
> The rest of the configurations, including sampling strategies and input resolution, follow the defaults used in InternVideo2.5-Chat.
>
> To better understand the reproducibility challenges you encountered, could you clarify whether you used the official InternVideo2.5-Chat implementation from Hugging Face and integrated our STAVE module as described for STAVEQ2 in our supplementary material? Additionally, did you incorporate InternVideo2.5+STAVE into the lmms-eval library (v0.3.0) for evaluation, as we did for STAVEQ2? Regarding the STAVE module, did you reinitialize the added temporal blocks, as this can impact training stability? Could you also specify which parts of the model were fine-tuned and the approximate number of parameters involved and GPU hours used for training? These details would help us identify any discrepancies in the setup and assist in resolving the issue.

---

### Official Review · Reviewer_rKct · 2025-06-29

**Clarity:** 2
**Significance:** 3
**Originality:** 3
**Rating:** 5
**Confidence:** 3

**Summary:**

This work focuses on improving the handling of temporal information by adopting stacked temporal attention in vision encoders. The motivation for breaking down the current Large Vision Language Model (LVLM) seems solid and reasonable to readers. However, it is somewhat challenging to fully understand why spatial and then stacked temporal works better than the spatiotemporal approach. It would be great if the author presented a bit more theoretical or empirical details.

**Questions:**

[Major]
I failed to catch why adopting 'stacked temporal attention' is the best approach over the rest. I understand that it could help the model handle the temporal dimension more efficiently, but I wonder whether it is the clear winner for the general cases.
1. It would be great if they could provide a bit more theoretical or empirical details. For example, in Table 1 in Supplementary Section B, I believe it would provide a clearer comparison if the table also included spatiotemporal information.
2. It would be more powerful if they provided a similar experiment over the pretrained InternVideo model (by replacing their spatiotemporal attention).
3. Similarly, it would be even more powerful if they performed the InternVideo experiment (= adding rows) in Tables 3 and 4.


[Minor]
1. Based on the code attached in the supplementary material, I believe the author will release the data, not just the attention implementation. However, I haven't seen a clear statement about data in their manuscript. It would be even better if they also mentioned the license along with the (curated/processed) datasets.

**Ethical Concerns:**

["NO or VERY MINOR ethics concerns only"]

**Final Justification:**

Thank the authors for their detailed response. I read the whole review and the rebuttal response fairly addressed my concerns, although some of them sound a bit indirect to me. For example, InternVideo2 + STA may introduce more parameters, and it would be much clearer if they could provide more apple-to-apple comparisons by replacing the spatiotemporal and introducing their STA with finetuning. Nonetheless, as I got understandable responses for most of the main questions, I would like to stick to my current score. I hope this rebuttal strengthens their manuscript.

**Limitations:**

yes

**Paper Formatting Concerns:**

I would like to leave only a few very marginal things.
1. The numerical citation is NOT a noun. There are a few numeric citation-only sentences.
2. The author did a great job of cleaning the citation, I guess. I can see a few items that you may want to double-check whether it is \inproceeding or not (e.g., [28]).

**Quality:**

3

**Strengths And Weaknesses:**

Strengths:
1. Great analysis of the current limitation on visual understanding of LVLM.
2. Following problem identification and their approach also sounds reasonable.
3. This manuscript compares their performance across multiple models, including some ablations.
4. Equations and figures are promptly used. It helped the reader understand very well.

Weaknesses:
1. It is somewhat challenging to clearly discern the author's approach, as it does not fully incorporate theoretical or empirical details explaining why this approach would perform better than spatiotemporal attention.
2. In line with the previous point, it is a bit unclear why InternVideo (which incorporates spatiotemporal attention) is not included in the main tables, Tables 3 and 4 (but is instead included in Table 2 only).
3. I failed to see the authors' plan for open-sourcing the data (and their license).

---

> ### Author Rebuttal · Authors · 2025-07-31
>
> Thank you for your detailed feedback and questions. We are encouraged by the positive assessment of our work, and we address each of your concerns below and provide clarifications, additional results, and plans for the camera-ready version.
>
> **W2 & Q3:** Absence of InternVideo2-Chat in Tables 3 and 4
>
> Thanks for pointing this out. This is due to a fundamental architectural limitation of the public InternVideo2-Chat model: it is restricted to processing a maximum of 8 input frames in each prompt. This is fine for SSv2 dataset which has video length of 2 to 6 seconds, but we cannot evaluate it on benchmarks that have longer video lengths like VideoMME (ranging from 11 seconds to over an hour). Therefore, a direct and fair comparison on those specific tables was not feasible. We will add a clear statement to the camera-ready version of the paper explaining this limitation. We will perform comparisons on suitable short-context benchmarks.
>
> **W1 & Q[Major]** Theoretical and empirical justification for Stacked Temporal Attention (STA)
>
> Thanks for these crucial questions. Our approach is grounded in empirical validation and clear architectural advantages. As demonstrated across our comprehensive results (Subsection 5.5, Table 4; Appendix F, Table 6; Rebuttal, Table 1), adding STA to vision encoders consistently improves performance, proving it is superior to relying on spatial-only attention.
>
> **Table 1**
> | Model                     | Comp. | Dir | Int. | Loc. | Seq. | Type | MVBench | VideoMME(wo) |
> | :-----------------------: | :---: | :--: | :--: | :--: | :--: | :--: | :-----: | :----------: |
> | VideoRoPE[1]              | 81.1  | 81.8 | 60.9 | 79.4 | 80.7 | 85.8 | 57.3    | 61.6         |
> | VideoRoPE+STAVE           | **81.9** | **82.9** | **61.8** | **79.9** | **81.3** | **86.3** | **59.2** | **62.5** |
> | InternVideo2.5-Chat[2]    | 91.3  | 88.7 | 82   | 84.8 | 84.7 | 91   | 75.7    | 65.1         |
> | InternVideo2.5-Chat+STAVE | **91.6** | **89.7** | **82.7** | **85.6** | **85.8** | **91.3** | **76.8** | **65.9** |
>
>
> To provide stronger evidence, we tested STA on the InternVideo2 vision encoder, which was explicitly trained on the Something-Something v2 (SSv2) dataset and held the previous state-of-the-art (SOTA) result. Our InternVideo2+STA model surpassed this powerful baseline and achieved a new SOTA on the SSv2 classification (Subsection 5.2, Table 2). This empirical evidence is critical, as it shows that dedicated temporal attention enhances the model’s ability to interpret temporal dynamics *even when added on top of an existing spatiotemporal attention mechanism*, enforcing more explicit modeling. An example attention map visualization of InternVideo2+STA is presented in Appendix E, Figure 1, which shows that our model allocates more attention to motion-relevant regions.
>
> Furthermore, the STA module is **additive**, preserving the powerful pre-trained spatial weights of the vision encoder. In contrast, converting a baseline to use spatiotemporal attention is a **destructive** process. For instance, modifying Qwen2-VL would require replacing 2D RoPE with 3D RoPE, which disrupts pre-trained weights and necessitates extensive retraining. Our approach elegantly bypasses this problem, demonstrating a clear advantage in performance and efficiency.
>
> Regarding whether STA is the "clear winner" in all general cases, our results confirm its superiority for tasks requiring fine-grained temporal reasoning.
>
> **Q1:** Adding spatiotemporal attention to Appendix B, Table 1
>
> Modifying Qwen2-VL’s spatial attention in its vision encoder to incorporate spatiotemporal attention involves replacing 2D Rotary Position Embedding (RoPE) with 3D RoPE. This change disrupts the pre-trained weights and requires extensive retraining. In contrast, the STA module is additive, allowing us to preserve the powerful pre-trained spatial weights of the vision encoder, resulting in a more robust training process which is unfortunately beyond the computation resource budget that is available to us. In Appendix B, Table 1, results for training STAVEQ2 and Qwen2-VL on SSV2-T10 dataset are presented. However, this dataset alone is insufficient for training the vision encoder to adapt to new position embeddings.
>
> **Q2** Experimenting over InternVideo2 by replacing their spatiotemporal attention
>
> Replacing the core spatiotemporal attention in the InternVideo2 vision foundation model with spatial-only attention would remove its primary mechanism for temporal reasoning, significantly limiting its capacity for video understanding. Instead, we chose to explore a more complementary direction by integrating our STA module into InternVideo2's existing spatiotemporal attention. The results of this experiment are presented in Subsection 5.2, Table 2.
>
> **W3 & Q[Minor]:** Plan for open-sourcing the data and their license
>
> Thanks for bringing this to our attention. We would like to clarify that all source datasets used are publicly available, and we have provided citations. Importantly, we will release our complete data processing pipeline and the implementation code for our models, as these have already been included in the supplementary materials.
>
> We hope these clarifications and new results have addressed your concerns. We are confident that these additions will significantly strengthen the paper.
>
> **References**
>
> [1] VideoRoPE: What Makes for Good Video Rotary Position Embedding?, Xilin Wei, Xiaoran Liu, Yuhang Zang, Xiaoyi Dong, Pan Zhang, Yuhang Cao, Jian Tong, Haodong Duan, Qipeng Guo, Jiaqi Wang, et al., International Conference on Machine Learning 2025.
>
> [2] InternVideo2.5: Empowering Video MLLMs with Long and Rich Context Modeling, Yi Wang, Xinhao Li, Ziang Yan, Yinan He, Jiashuo Yu, Xiangyu Zeng, Chenting Wang, Cheng Ma, Hong Huang, Jing Gao, and Meng Dou, arXiv preprint arXiv:2501.12386 2025.

---

> > ### Comment · Reviewer_rKct · 2025-08-05
> > **Thank the authors for their detailed response**
> >
> > Thank the authors for their detailed response. I read the whole review and the rebuttal response fairly addressed my concerns, although some of them sound a bit indirect to me. For example, InternVideo2 + STA may introduce more parameters, and it would be much clearer if they could provide more apple-to-apple comparisons by replacing the spatiotemporal and introducing their STA with finetuning. Nonetheless, as I got understandable responses for most of the main questions, I would like to stick to my current score. I hope this rebuttal strengthens their manuscript.

---

> > > ### Author Response · Authors · 2025-08-06
> > >
> > > Thank you for your thoughtful and detailed feedback throughout the review process and for your positive assessment of our work. We greatly appreciate your time and careful consideration, which encouraged us to further strengthen our manuscript.
> > >
> > > Regarding the parameter count for InternVideo2+STA, we clarify that InternVideo2 1B+STA has a total of approximately 1.3B parameters. However, the performance improvement is not solely due to the increased parameters. As shown in Section 5.3, Table 2, InternVideo2 1B+STA not only surpasses the base InternVideo2 1B but also outperforms the larger InternVideo2 6B model, which has significantly more parameters. This suggests that the STA module’s dedicated temporal attention mechanism contributes to the performance gains, beyond the effect of additional parameters.

---

> ### Comment · Area_Chair_ACJF · 2025-08-05
>
> Dear Reviewer,
>
> Thank you for your dedicated efforts in reviewing this paper. We are currently in the reviewer-author discussion phase, but we have not yet seen your engagement.
>
> This year's Responsible Reviewing initiative requires all reviewers to communicate with authors during this period, emphasizing that ghosting is not acceptable. We kindly ask that you reply and engage with the authors. Please note that participation in discussions with authors is mandatory before submitting "Mandatory Acknowledgement," as submitting it without any engagement is not permitted in this review cycle.
>
> Best,
> AC

---

### Official Review · Reviewer_q8eF · 2025-07-01

**Clarity:** 3
**Significance:** 3
**Originality:** 3
**Rating:** 5
**Confidence:** 4

**Summary:**

This paper presents STAVEQ2, a novel video language model (Video-LLM) architecture that explicitly incorporates stacked temporal attention modules within the vision encoder. The motivation is that existing Video-LLMs struggle with fine-grained temporal reasoning, particularly in action recognition tasks requiring understanding of motion direction or temporal order. The authors demonstrate limitations of existing Video-LLMs (e.g., Qwen2-VL, InternVideo2) on temporally challenging benchmarks. They propose STAVEQ2, which adds efficient temporal attention after spatial attention blocks in the vision encoder. Showing performance gains over strong baselines on multiple video understanding benchmarks, including SSv2, MVBench, VITATECS, and Video-MME.The authors will release code (as a supplement) and propose both action recognition and visual similarity tasks to evaluate temporal modeling.

**Questions:**

Could the authors provide qualitative visualizations of what the temporal attention captures (e.g., token-level attention across frames)? This would strengthen interpretability.

How would STAVEQ2 scale to long-form video (e.g., >30 seconds)? Are the temporal attention blocks placed uniformly across layers, or could they be adaptively placed?

How does STAVEQ2 compare to alternative temporal modeling strategies such as divided space-time attention or hierarchical token merging? Are there trade-offs?

Can the same architecture generalize well to tasks like video captioning or moment retrieval? Would different tuning be needed?

Could the authors isolate the effect of using 1D RoPE vs. 2D or the inclusion of LoRA adapters during training?

**Ethical Concerns:**

["NO or VERY MINOR ethics concerns only"]

**Final Justification:**

Considering the rebuttal and the reading of the other reviews, I am happy to recommend the same rating as before

**Limitations:**

The paper briefly discusses limitations related to computational constraints and model scaling (Section 6), which is fair. However, it would be helpful to examine whether temporal modeling introduces brittleness to frame sampling or frame rate changes. Any risks of overfitting to short clip-based benchmarks like SSv2.Potential societal impacts of enhanced video understanding (e.g., surveillance, deepfake detection), if relevant.

**Quality:**

3

**Strengths And Weaknesses:**

Strengths:

The problem is well-identified — existing Video-LLMs poorly capture temporal dynamics.

Incorporating stacked temporal attention directly into the vision encoder is a clear architectural improvement, conceptually simple yet effective.

The paper is evaluated on multiple benchmarks, including both established (SSv2, MVBench) and diagnostic (VITATECS) benchmarks, showing consistent gains.

Ablation and component study: Shows substantial gains over baselines like InternVideo2 with fewer parameters, especially on temporally mirrored action classes.

The design is lightweight, using fewer attention heads and RoPE adaptations, making it appealing for real-world use, with Table 4 showing performance across a range of model sizes

Supplementary material includes code and implementation details.

Weaknesses:

While effective, the idea of adding temporal attention blocks is a relatively incremental change and follows known design principles from ViTs.

The paper could benefit from further analysis of why certain temporal attention blocks are more effective (e.g., visualization of learned temporal relations).

STAVEQ2 is not evaluated on long video sequences beyond the SSv2/MVBench scale, which limits its claims for “temporal reasoning at scale.”

 There is no mention of confidence intervals or variance across runs. Gains are reported as absolute.

---

> ### Author Rebuttal · Authors · 2025-07-31
>
> We thank you for your thorough and positive review, as well as the constructive suggestions. Below, we address the identified weaknesses and answer the questions in detail.
>
> **W1**: Incremental change
>
> Architecturally, our Stacked Temporal Attention (STA) blocks are designed to be additive and parameter-efficient. This preserves the powerful pre-trained spatial weights of the vision encoder and ensures training stability, making it a more viable approach than training a model from scratch or performing a destructive architectural change. Our main focus in this work is to *systematically demonstrate the necessity of temporal attention in Video-LLMs*, where it has been largely overlooked or dismissed by recent architectural trends. For instance, Qwen2-VL has only focused on developing Multimodal Rotary Position Embedding for temporal understanding, while relying solely on spatial attention without explicit temporal modeling in the vision encoder, as stated in their paper [6].
>
> Our comprehensive results, presented in Subsection 5.5, Table 4, and Appendix Section F, Table 6 of our paper, and further supported by Table 2 in our rebuttal, directly challenge this design paradigm. These results unequivocally show that incorporating explicit temporal attention mechanisms, like our Stacked Temporal Attention, leads to substantial performance improvements on benchmarks that specifically require robust temporal understanding. Table 1 of rebuttal also shows that newer models like **VideoRoPE** and **InternVideo2.5** which focus on different approaches for advancing temporal understanding, can still benefit from this effective method. We will emphasize this positioning more clearly in the camera-ready version.
>
> Another compelling proof of this is our experiment on the InternVideo2 vision encoder—the previous state-of-the-art (SOTA) model on SSv2, which already incorporates spatiotemporal attention. Our InternVideo2+STA model surpassed this powerful baseline and achieved a new SOTA on SSv2 classification (Subsection 5.2, Table 2). This critical result shows that dedicated, explicit temporal modeling enhances performance even on top of an existing spatiotemporal mechanism.
>
> This is also reflected in the positive feedback from other **reviewer tYNw**, describing our work as *"a novel approach that effectively addresses the limitations of current Video-LLMs in temporal understanding."*
>
> **W2 & Q1**: Visualization of temporal attentions
>
> We agree that interpretability is an important aspect. To address this, we provide an attention map visualization in **Appendix E, Figure 1** that compares InternVideo2 with our InternVideo2+STA. This visualization demonstrates that our model more effectively allocates attention to motion-relevant regions. While new figures cannot be included in the rebuttal itself, we will expand this analysis with additional examples in the camera-ready version to further strengthen the interpretability of our approach.
>
>
> **W3 & Q2**: Evaluation on long video sequences
>
> Thank you for this question regarding scalability to longer video clips. Our model's scalability is demonstrated on the VideoMME benchmark, where it achieves consistent gains across videos ranging from **11 seconds to over an hour**. The current limitation on maximum video length is not inherent to our design but is imposed by the base Qwen2-VL architecture's hard context limit (approximately 760 frames). In principle, our Stacked Temporal Attention blocks are designed to scale efficiently with input length.
>
> **W4**: Confidence interval and variance
>
> Thank you for this question. Our evaluation protocol follows the established standard for large-scale Video-LLM benchmarking, and is consistent with the evaluation practices of recent leading works in the field, including Qwen2-VL. We report results from a single training run due to the prohibitive computational cost. A single run for models of this scale requires hundreds of GPU-hours, making multiple independent runs infeasible.
>
> **Q2**: Temporal attention block placement
>
> In our current design, we add one temporal attention layer to each transformer block.
>
> To assess the impact of STA placement, we conducted two additional ablation studies, in addition to those in Appendix B, that varied the number and placement of temporal blocks within the transformer blocks. The results are summarized in Table 1 of our rebuttal, illustrating how different configurations affect the model’s accuracy. As shown, reducing the number of temporal blocks from 32 to 16 seems to decrease performance. Additionally, the placement of 16 temporal blocks—whether distributed uniformly or concentrated in the early layers of the vision encoder—does not result in significant differences in performance. We will add these experiments to Appendix B, Table 1, in our camera-ready version.
>
> **Table 1**
>
> | Model        | Number of Temporal Blocks | Temporal Blocks Placement | Accuracy (%) |
> | :----------: | :-----------------------: | :-----------------------: | :----------: |
> | STAVEQ2 2B| 32 | All Blocks | 76.04 |
> | STAVEQ2 2B| 16                        | Uniform                   | 74.73        |
> | STAVEQ2 2B| 16                        | First Blocks              | 74.97        |
>
>
>
>
> **Q3**: Comparison to other strategies
>
> While other lines of research also aim to improve temporal awareness—such as hierarchical token merging in InternVideo2.5-Chat [3] or advanced positional encodings in VideoRoPE [2]—we view these methods as orthogonal and complementary to our approach. To validate this, we conducted new experiments applying our method to these strong baselines. As presented in Table 2 of our rebuttal, we observed consistent performance gains across all models, demonstrating the synergistic value of our approach.
>
> **Table 2**
>
> | Model                     | Comp. | Dir | Int. | Loc. | Seq. | Type | MVBench | VideoMME(wo) |
> | :-----------------------: | :---: | :--: | :--: | :--: | :--: | :--: | :-----: | :----------: |
> | VideoRoPE [1]              | 81.1  | 81.8 | 60.9 | 79.4 | 80.7 | 85.8 | 57.3    | 61.6         |
> | VideoRoPE+STAVE           | **81.9** | **82.9** | **61.8** | **79.9** | **81.3** | **86.3** | **59.2** | **62.5** |
> | InternVideo2.5-Chat [2]    | 91.3  | 88.7 | 82   | 84.8 | 84.7 | 91   | 75.7    | 65.1         |
> | InternVideo2.5-Chat+STAVE | **91.6** | **89.7** | **82.7** | **85.6** | **85.8** | **91.3** | **76.8** | **65.9** |
>
>
> **Q4**: Generalization to other tasks
>
> To assess generalization to other tasks such as video captioning, we evaluated our STAVEQ2 7B model against its Qwen2-VL 7B baseline on the challenging CaReBench benchmark [4]. CaReBench is specifically designed to test a model's perceptual accuracy by evaluating it across two categories, 'Action' and 'Object', using F1, Precision, and Recall scores to ensure factual correctness and mitigate hallucination. Our results in Table 3 demonstrate that STAVEQ2 exhibits stronger captioning abilities and improves upon the base model without any caption-specific fine-tuning.
>
>
> **Table 3**
>
> | Model               | A-F1  | A-Percision | A-Recall | O-F1  | O-Percision | O-Recall |
> | :-----------------: | :---: | :---------: | :------: | :---: | :---------: | :------: |
> | Qwen2-VL 7B         | 0.34 | 0.47       | 0.27    | 0.34 | 0.55        | 0.25    |
> | STAVEQ2 7B (Ours)| **0.41** | **0.49** | **0.36** | **0.43** | **0.61** | **0.33** |
>
>
> For moment retrieval, task-specific tuning would likely be required. The attention mechanism itself is general and can integrate into these models.
>
> **Q5**: Isolating the effect of using 1D RoPE vs. 2D or the inclusion of LoRA adapters during training
>
> 1. In our architecture, 2D RoPE is applied to the spatial attention layers and 1D RoPE is applied to our separate temporal attention layers. As these embeddings operate on orthogonal dimensions (space and time), a direct "A vs. B" ablation is not straightforward. We would be grateful if you could clarify the specific experiment they envision so we can provide a more detailed response.
>
> 2. We isolated the effect of LoRA through a direct comparison on the SSv2 benchmark. We first confirmed that the SOTA InternVideo2 baseline could not be improved further with either LoRA or full fine-tuning. However, when we introduced our architectural change by adding STA (InternVideo2+STA), the model surpassed this baseline and achieved a new SOTA (Subsection 5.2, Table 2). This comparison clearly demonstrates that *the significant performance gain is attributable to our STA module, not the training method.*
>
> **References**
>
> [1] Qwen2-VL: Enhancing Vision-Language Model's Perception of the World at Any Resolution, Peng Wang et. al., arXiv preprint arXiv:2409.12191 2024.
>
> [2] VideoRoPE: What Makes for Good Video Rotary Position Embedding?, Xilin Wei et al., International Conference on Machine Learning 2025.
>
> [3] InternVideo2.5: Empowering Video MLLMs with Long and Rich Context Modeling, Yi Wang et. al., arXiv preprint arXiv:2501.12386 2025
>
> [4] CaReBench: A Fine-Grained Benchmark for Video Captioning and Retrieval, Yifan Xu et. al., arXiv preprint arXiv:2501.00513 2025.

---

> ### Author Response · Authors · 2025-08-03
>
> Dear Reviewer,
>
> Thank you for your initial feedback. We have submitted a detailed rebuttal addressing your comments and would greatly appreciate it if you could review our responses and share any further questions or thoughts. Your engagement is important for clarifying key points, and we’re happy to provide additional details if needed.
>
> Thank you for your time and effort!

---

> ### Comment · Area_Chair_ACJF · 2025-08-05
>
> Dear Reviewer,
>
> Thank you for your dedicated efforts in reviewing this paper. We are currently in the reviewer-author discussion phase, but we have not yet seen your engagement.
>
> This year's Responsible Reviewing initiative requires all reviewers to communicate with authors during this period, emphasizing that ghosting authors is not acceptable. We kindly ask that you reply and engage with the authors.
>
> Best,
> AC

---

> > ### Comment · Reviewer_q8eF · 2025-08-06
> >
> > The response to my queries and the reading of the other reviews including the reject one, have led me to maintain my rating. In particular their response to the incremental contribution concern  has reassured me

---

> > > ### Author Response · Authors · 2025-08-06
> > >
> > > Thank you for your thoughtful and detailed feedback throughout the review process and for your positive assessment of our work. We greatly appreciate your time and careful consideration, which encouraged us to further strengthen our manuscript.

---

### Official Review · Reviewer_58rP · 2025-07-04

**Clarity:** 3
**Significance:** 3
**Originality:** 1
**Rating:** 4
**Confidence:** 5

**Summary:**

The paper proposes Stacked Temporal Attention to be a mechanism to improve the temporal awareness of the vision encoder used in VLMs. Specifically, it applies the attention across tokens on the same spatial location across times, in addition to the per-frame attention. It created a subset of SSv2 and show that the proposed modification improve the performance on this specific set covering action classes that particularly require temporal awareness. Also, the authors benchmark the proposed approach on standard video understanding benchmarks like MVBench, VideoMME etc.

**Questions:**

- For the fine-tuning results in Figure 2c, what's the size of the held-out set on SSv2-T10?
- Typos: L201, S^{(m)}1, i -> S^{(m)}_{1, i}
- Typos: Eq 7, "Z^{(m)}" -> "S^{(m)}", according to the texts in L206

**Ethical Concerns:**

["NO or VERY MINOR ethics concerns only"]

**Final Justification:**

I thank the reviewers for their efforts in the rebuttal. I appreciate the added baselines and discussions on VideoRoPE and InternVideo2.5-Chat. To reiterate, I never argued against the importance of incorporating temporal attention in vision encoders for LLM, what I concerned about is the fact that there has been prior work studying the interplay of spatial and temporal attention (albeit for video classification task) and I think this paper has not addressed that fact enough. That said, I do agree that the current trend of designing *vision encoder for LLM* is not putting enough focus to incorporate temporal attention and this paper could serve to raise attention on that. Given that, I will raise my rating to borderline accept the paper into the conference.

**Limitations:**

Yes

**Quality:**

2

**Strengths And Weaknesses:**

- Table 1, the low performance with visual prompting is not surprising, as most VLM models are not trained with enough number of multi-turn visual prompting data. Thus IMO this visual prompting experiment does not add much information upon the zero-shot results.
- The proposed modification could in principle be applied to other vision-encoder/VLM, beyond Qwen2 models. It would be more useful if the authors could show some results generalizing it beyond Qwen2 models.
- There is limited novelty in this paper, as the proposed "Stacked Temporal Attention" has been explored before (e.g. in [1]) and studied as one variant of the space-time transformers.
- There is another line of work aiming to improve the temporal awareness via better encoding (e.g. VideoRoPE [2]), which should be included as the baseline for this work.

[1] Is Space-Time Attention All You Need for Video Understanding?, Gedas Bertasius, Heng Wang, Lorenzo Torresani
[2] VideoRoPE: What Makes for Good Video Rotary Position Embedding?, Wei et al. ICML 2025

---

> ### Author Rebuttal · Authors · 2025-07-31
>
> We thank you for the detailed comments. Below, we address each concern and provide clarifications, results, and plans for the camera-ready version.
>
> **W1:** Visual prompting experiments
>
> Our main goal here is to properly analyze current state-of-the-art foundational models for their basic temporal understanding capabilities. Usually when some capabilities are missing in a family of models one of the important baselines and ablations is to try to simply train the models to achieve that capability. Apart from convectional training approaches, another way to teach a LLM a new capability/skill is to use few shots samples and in context learning. Some recent studies [1,2] show that multimodal LLMs benefit from in-context learning without further training needed. To this end, we also included a visual prompting ablation study to properly analyze the temporal understanding capabilities of current state-of-the-art foundational video LLMs. Our experiment illustrates a critical point: *even with proper visual prompting, current video LLMs fail to capture fine-grained temporal dependencies*, which are essential for accurate video understanding. We conduct all these experiments to properly identify the current capabilities of state-of-the-art foundational video LLMs and identify the problem, *as mentioned by all of the other reviewers as our strengths:*
>
> *"The problem is well-identified" (**reviewer q8eF**)*
>
> *"Great analysis of the current limitation on visual understanding of LVLM." and "Following problem identification and their approach also sounds reasonable." (**reviewer rKct**)*
>
> *"The authors conduct a thorough analysis of the limitations of current Video-LLMs, providing valuable insights into their shortcomings in temporal reasoning. This analysis motivates the proposed approach and underscores the importance of temporal modeling in video understanding tasks." (**reviewer tYNw**)*
>
>
> **W2:** Qwen2-VL Models
>
> Thanks for raising this point. We want to clarify that our approach is not limited to Qwen2-VL. In the submitted version, we evaluated **STAVEQ2** (Subsection 5.5, Table 4), which builds upon **Qwen2-VL**, and **STAVEQ2.5**  (Appendix F, Table 6), which builds upon **Qwen2.5-VL**. For this rebuttal, we further applied our method to other recent and highly competitive Video-LLMs, **VideoRoPE [3]** and **InternVideo2.5-Chat [4]**, which also aim to improve temporal awareness. We observed consistent improvements across all these multimodal Video-LLMs. Additionally, we demonstrated that applying Stacked Temporal Attention to the **InternVideo2** vision foundation model achieves a new **state-of-the-art on the SSv2** dataset. This broad evaluation demonstrates strong generalization across different model families and design paradigms, validating that our proposed mechanism is widely applicable. We will include these additional results in the camera-ready version.
>
> Broader evaluation of all models would be ideal but is infeasible due to limited GPU resources.
>
> Here are the results for the newly experimented models:
>
>
> **Table 1**
>
> | Model                     | VITATECS                |           |           |           |           |           | MVBench | VideoMME(wo) |
> |:-------------------------:|:-----------------------:|:---------:|:---------:|:---------:|:---------:|:---------:|:-------:|:------------:|
> |                           | Comp.                   | Dir       | Int.      | Loc.      | Seq.      | Type      |         |              |
> | VideoRoPE[3]                 | 81.1                    | 81.8      | 60.9      | 79.4      | 80.7      | 85.8      | 57.3    | 61.7         |
> | VideoRoPE+STAVE           | **81.9**                    | **82.9**      | **61.8**      | **79.9**      | **81.3**      | **86.3**      | **59.2**    | **62.5**         |
> | InternVideo2.5-Chat[4]       | 91.3                    | 88.7      | 82        | 84.8      | 84.7      | 91        | 75.7    | 65.1         |
> | InternVideo2.5-Chat+STAVE | **91.6**                    | **89.7**      | **82.7**      | **85.6**      | **85.8**      | **91.3**      | **76.8**    | **65.9**         |
>
>
>
> **W3:** Novelty
>
> The mentioned prior work [5] primarily focuses on end-to-end video classification models. However, in this work we focus on the possibility of improving temporal understanding capabilities of the pretrained vision encoder models that output vision tokens to a generative LLM. Architecturally, our Stacked Temporal Attention (STA) blocks are designed to be additive and parameter-efficient. This preserves the powerful pre-trained spatial weights of the vision encoder and ensures training stability, making it a more viable approach than training a model from scratch or performing a destructive architectural change. We *systematically demonstrate the necessity of having temporal attentions in Video-LLMs*, where it has been largely overlooked or dismissed by recent architectural trends. For instance, Qwen2-VL has only focused on developing Multimodal Rotary Position Embedding for temporal understanding, while relying solely on spatial attention without explicit temporal modeling in the vision encoder, as stated in their paper [6].
>
> Our comprehensive results, presented in Subsection 5.5, Table 4, and Appendix Section F, Table 6 of our paper, and further supported by Table 1 in our rebuttal, directly *challenge this design paradigm*. These results unequivocally show that incorporating explicit temporal attention mechanisms, like our Stacked Temporal Attention, leads to substantial performance improvements on benchmarks that specifically require robust temporal understanding. We will emphasize this positioning more clearly in the camera-ready version.
>
> Another compelling proof of this is our experiment on the InternVideo2 vision encoder—the previous state-of-the-art (SOTA) model on SSv2, which already incorporates spatiotemporal attention. Our InternVideo2+STA model surpassed this powerful baseline and achieved a new SOTA on SSv2 classification (Subsection 5.2, Table 2). This critical result shows that dedicated, explicit temporal modeling enhances performance even on top of an existing spatiotemporal mechanism.
>
> This is also reflected in the positive feedback from other reviewers: **Reviewer q8eF** notably described our method as *"conceptually simple, yet effective,"* calling it a *"clear architectural improvement,"* while **reviewer tYNw** described our work as *"a novel approach that effectively addresses the limitations of current Video-LLMs in temporal understanding."*
>
>
> **W4:** VideoRoPE
>
> Although there are other lines of research that aim to improve the temporal understanding of Video-LLMs, such as VideoRoPE [3], they are orthogonal to our approach. To better study this, we have conducted additional experiments incorporating VideoRoPE as a baseline.
>
> It's important to clarify the distinct focuses of our approaches: VideoRoPE primarily addresses the positional embedding of output feature tokens fed to the LLM, whereas our Stacked Temporal Attention (STA) focuses on enhancing token-level temporal interactions directly within the vision encoder. These are complementary improvements, and our new experiments confirm this: applying STA to VideoRoPE consistently improves its performance across various benchmarks (see Table 1 of our rebuttal). We will include these results and highlight this synergistic relationship in the camera-ready version of the paper.
>
>
>
> **Q1:** SSv2-T10 info
>
> For evaluating both the **base** and **STA-added models** on SSv2-T10, we used a subset of **2,116** samples with this distribution:
>
>  - Pulling [something] from left to right – 14.68%
>
> - Pulling [something] from right to left – 14.97%
>
> - [Something] falling like a rock – 13.12%
>
> - Picking [something] up – 9.26%
>
> - Throwing [something] in the air and letting it fall – 7.31%
>
> - Throwing [something] in the air and catching it – 8.90%
>
> - Moving [something] away from [something] – 8.61%
>
> - Moving [something] closer to [something] – 8.55%
>
> - Rolling [something] on a flat surface – 11.85%
>
> - Poking a stack of [something] so the stack collapses – 2.74%
>
> We will include the exact details in the appendix/supplementary document of our camera ready version.
>
> **Typos**
>
> Thanks for pointing out **the typo in line 201**. We will fix this for the camera-ready version.
>
> However, there is **no typo in Equation 7**, and it is indeed correct. As shown below, our architecture incorporates residual connections within each attention block, taking the input to that block and adding it to its output. We will ensure the accompanying text in the camera-ready version clearly reflects this. We appreciate you highlighting these details.
>
> Here are the equations for clarity:
>
> Spatial attention block:
>
> $$
> S^{(m)} = \text{(concatenation of\ } S^{(m)}_t \text{\ vectors along the temporal dimension)} + \text{(the residual connection from\ } X^{(m-1)})
> $$
>
> Temporal attention block:
>
> $$
> Z^{(m)} = \text{(concatenation of\ }  Z^{(m)}_i  \text{\ vectors along the spatial dimension)} +  \text{(the residual connection from\ }  S^{(m)})
> $$
>
> Final block:
>
> $$
> X^{(m)} = MLP(LN(Z^{(m)})) + Z^{(m)}
> $$
>
> **References**
>
> [1] In-context learning enables multimodal large language models to classify cancer pathology images, Dyke Ferber et. al., Nature 2024.
>
> [2] MM-Narrator: Narrating Long-form Videos with Multimodal In-Context Learning, Chaoyi Zhang et. al., CVPR 2024.
>
> [3] VideoRoPE: What Makes for Good Video Rotary Position Embedding?, Xilin Wei et. al., ICML 2025.
>
> [4] InternVideo2.5: Empowering Video MLLMs with Long and Rich Context Modeling, Yi Wang et. al. arXiv preprint arXiv:2501.12386 2025.
>
> [5] Is Space-Time Attention All You Need for Video Understanding?, Gedas Bertasius et. al., ICML 2021.
>
> [6] Qwen2-VL: Enhancing Vision-Language Model's Perception of the World at Any Resolution, Peng Wang et. al. arXiv preprint arXiv:2409.12191 2024.

---

> ### Author Response · Authors · 2025-08-03
>
> Dear Reviewer,
>
> Thank you for your initial feedback. We have submitted a detailed rebuttal addressing your comments and would greatly appreciate it if you could review our responses and share any further questions or thoughts. Your engagement is important for clarifying key points, and we’re happy to provide additional details if needed.
>
> Thank you for your time and effort!

---

> ### Comment · Area_Chair_ACJF · 2025-08-05
>
> Dear Reviewer,
>
> Thank you for your dedicated efforts in reviewing this paper. We are currently in the reviewer-author discussion phase, but we have not yet seen your engagement.
>
> This year's Responsible Reviewing initiative requires all reviewers to communicate with authors during this period, emphasizing that ghosting authors is not acceptable. We kindly ask that you reply and engage with the authors.
>
> Best,
> AC

---

### Author Response · Authors · 2025-08-01

We appreciate the reviewers' recognition of the novelty of our work and their constructive feedback. We responded to the comments and added the following experiments and analyses:

- In addition to the models in our submission, we applied our method to two recent Video-LLMs, VideoRoPE [1] and InternVideo2.5 [2], observing consistent performance improvements across all benchmarks. This addresses **reviewer 58rP** and **reviewer tYNw**'s concerns regarding additional baselines, and shows our approach's robustness to architectural variations and effectiveness even when layered on top of other temporal modeling strategies (**reviewer q8eF, reviewer 58rP**), as summarized in Table 1:

| Model | VITATECS | | | | | | MVBench | VideoMME (wo) |
|:---:|:---:|:---:|:---:|:---:|:---:|:---:|:---:|:---:|
| | Comp. | Dir | Int | Loc. | Seq. | Type | | |
| VídeoRoPE[3] | 81.1 | 81.8 | 60.9 | 79.4 | 80.7 | 85.8 | 57.3 | 61.7 |
| VídeoRoPE+STAVE | **81.9** | **82.9** | **61.8** | **79.9** | **81.3** | **86.3** | **59.2** | **62.5** |
| InternVideo2.5-Chat[4] | 91.3 | 88.7 | 82 | 84.8 | 84.7 | 91 | 75.7 | 65.1 |
| InternVideo2.5-Chat+STAVE | **91.6** | **89.7** | **82.7** | **85.6** | **85.8** | **91.3** | **76.8** | **65.9** |

- In response to **reviewer q8eF**'s interest in generalization to other tasks, we benchmarked our STAVEQ2 model on the challenging CaReBench [3] video captioning benchmark, confirming consistent performance improvements over its baseline, as shown in Table 2:

    | Model               | A-F1  | A-Percision | A-Recall | O-F1  | O-Percision | O-Recall |
    | :-----------------: | :---: | :---------: | :------: | :---: | :---------: | :------: |
    | Qwen2-VL 7B         | 0.34 | 0.47       | 0.27    | 0.34 | 0.55        | 0.25    |
    | STAVEQ2 7B (Ours)| **0.41** | **0.49** | **0.36** | **0.43** | **0.61** | **0.33** |


- To address questions from **reviewer tYNw** and **reviewer q8eF** about the placement of temporal blocks, we conducted new ablation studies on STA depth and positioning to provide further architectural insights. Due to space constraints, details for this experiment are given in our responses to **reviewer tYNw** and **reviewer q8eF**.



Current Video-LLMs face challenges in modeling temporal dynamics in videos. We contribute to addressing these challenges via:
- Presenting a novel, thorough analysis that reveals the specific limitations of recent Video-LLMs in fine-grained temporal tasks (Section 3 of Paper).

- Proposing STAVEQ2, our Video-LLM equipped with improved temporal understanding via Stacked Temporal Attention blocks (Section 4 of Paper).

- Applying our approach to multiple state-of-the-art models, including those that use different temporal enhancement methods. This resulted in achieving new state-of-the-art performance on the SSv2 dataset and, to the best of our knowledge, on MVBench for models at the ~7B parameter scale (Section 5 of Paper and Rebuttal).

- Demonstrating the generalization of STAVEQ2 beyond VQA by achieving strong performance improvement on the CaReBench video captioning benchmark (Rebuttal).

The reviewers also acknowledged our contributions and strengths. They agreed that the paper tackles an important and well-defined problem—that current Video-LLMs struggle with fine-grained temporal understanding, particularly in tasks involving temporal order **(q8eF, tYNw)**. Our introduction of stacked temporal attention modules was noted as an effective architectural enhancement **(q8eF, tYNw)**, and the motivation and overall approach were regarded as clear and well-reasoned **(rKct, tYNw)**. The experimental validation was recognized as robust, demonstrating consistent improvements across multiple benchmarks, including MVBench, VITATECS, and Video-MME **(q8eF, tYNw)**. Reviewers specifically noted the achievement of state-of-the-art results on SSv2 action recognition **(tYNw)**. Furthermore, our comprehensive ablation studies and strong baseline comparisons were seen as reinforcing the effectiveness of our method **(q8eF, rKct, tYNw)**. The design was also appreciated for being parameter-efficient, making it practical for real-world applications **(q8eF, tYNw)**. Additionally, the clarity of presentation, supported by well-crafted equations and figures, was commended **(rKct)**.

In summary, our additional experiments during the rebuttal period have: (1) incorporated two powerful new baselines into our evaluation, which in turn confirmed the synergistic value and robustness of our approach when combined with other temporal enhancement methods; (2) demonstrated its wide applicability to other tasks such as video captioning; and (3) provided deeper insights through new ablations on the placement of STA blocks.

We are carefully addressing all feedback and will incorporate these updates and new results into the camera-ready version.

References are included in the following reply.

---

> ### Author Response · Authors · 2025-08-01
>
> **References**
>
> [1] VideoRoPE: What Makes for Good Video Rotary Position Embedding?, Xilin Wei et. al., ICML 2025.
>
> [2] InternVideo2.5: Empowering Video MLLMs with Long and Rich Context Modeling, Yi Wang et. al. arXiv preprint arXiv:2501.12386 2025.
>
> [3] CaReBench: A Fine-Grained Benchmark for Video Captioning and Retrieval, Yifan Xu et. al., arXiv preprint arXiv:2501.00513 2025.

---

### Note · Authors · 2025-08-15

Dear Area Chair and Reviewers,

Thank you for the opportunity to provide these final remarks. We appreciate the valuable feedback and discussion.

We are pleased that reviewers **q8eF** and **rKct** maintained their “Accept” recommendations and confirmed that our rebuttal resolved their concerns. Reviewer **q8eF** explicitly stated that, even after reading the concerns and discussion of another reviewer who had given a “Reject” score, their positive view of our work remained unchanged.

We diligently addressed all other points raised. In response to reviewer **58rP**, we implemented two additional baselines (including one specifically requested) and provided a detailed defense on the novelty of our work. We note that reviewer **q8eF**, who initially shared a similar concern, confirmed that our rebuttal satisfactorily resolved the issue for them.

For reviewer **tYNw**, who raised new questions during the discussion period, we also offered prompt clarifications. Specifically, we explained that the perceived dataset size discrepancy was due to an **incorrect calculation in their comment** involving overlapping subsets (our original numbers were correct); shared configurations and offered direct support to reproduce a post-submission experiment (as code cannot be uploaded during rebuttal); and confirmed that the baseline results in Table 4 of our paper were sourced from their original papers. As the discussion period concluded before we received a reply, we hope these clarifications resolve the final questions.

We stand by our work’s contributions. Our method, Stacked Temporal Attention (STA), directly targets the critical challenge of temporal understanding in Video-LLMs. It is a lightweight module that consistently improves diverse models, sets **a new state-of-the-art on SSv2 dataset and benchmarks like MVBench for ~7B models**, and its benefits generalize to other tasks like video captioning.

Thank you for your time and careful consideration.

Best regards,

Authors

---

### Decision · Program_Chairs · 2025-09-17

**Decision:**

Accept (poster)

**Comment:**

The final review scores were two "Accepts," one "Borderline Accept," and one "Borderline Reject."

All three positive reviewers explicitly stated that the author rebuttal sufficiently addressed their concerns. One reviewer subsequently raised their rating, and another maintained their stance for acceptance even after considering the negative review.

The negative reviewer, however, chose to keep their original rating following the post-rebuttal discussion. The main points of concern raised were: 1) the use of a different FPS in some baseline models, 2) minor calculation errors, and 3) the reviewer's failure to reproduce some results.

The AC has evaluated these concerns and finds them to be minor factors for the following reasons:
- Fair Comparison: The primary baseline model, which the proposed model is built upon and is its best-performing competitor, uses the same FPS, ensuring a fair comparison.
- Minor Mismatch: The calculation mismatch in the dataset is a minor error that does not negate the paper's overall validity.
- Reproduction Issues: The reproduction failure occurred on one setup with a single model, which the authors were only asked to include during the rebuttal period. Furthermore, the reviewer did not use the authors' code but their own reproduction efforts. This failure cannot negate the paper's overall validity.

Given the overall positive consensus and the minor nature of the concerns raised by the dissenting reviewer, the AC believes the paper meets the NeurIPS standard and recommends acceptance.